**A PORTRAIT OF CENTRAL ITALY'S GEOLOGY THROUGH GIOTTO'S PAINTINGS AND ITS POSSIBLE CULTURAL IMPLICATIONS**

Ann C. Pizzorusso

Independent Geologist, 511 Avenue of the Americas, New York, New York 10011 USA

*Correspondence to:* Ann C. Pizzorusso (tweetingdavinci@gmail.com)

**Abstract.** Central Italy has some of the most complex geology in the world. In the midst of this inscrutable territory, two people emerged--St. Francis and Giotto--they would ultimately change the history of ecology, religion and art by extoling the landscapes and geology of this region.

From Antiquity to the Middle Ages, man had a conflictual relationship with nature, seeing it as representing either divine or satanic forces. On the vanguard of a change in perspective toward the natural world was St. Francis of Assisi (c.1181-1226) who is now, thanks to his pioneering work, patron of ecology. He set forth the revolutionary philosophy that the Earth and all living creatures should be respected as creations of the Almighty.

St. Francis' affinity for the environment influenced the artist Giotto (c.1270-1337) who revolutionized art history by including natural elements in his religious works. By taking sacred images away from Heaven and placing them in an earthly landscape, he separated them definitively from their abstract, unapproachable representation in Byzantine art. Giotto's works are distinctive because they portray daily life as blessed, thus demonstrating that the difference between the sacred and profane is minimal.

Disseminating the new ideas of St. Francis visually was very effective, as the general populace was illiterate. Seeing frescoes reflecting their everyday lives in landscapes that were familiar, changed their way of thinking. The trees, plants, animals and rocky landscapes were suddenly perceived as gifts from the Creator to be used, enjoyed and respected. Further, Giotto recognized that the variety of dramatic landscapes would provide spectacular visual interest in the works. By including the striking landforms of central Italy, and portraying them accurately, Giotto allows us the opportunity to identify the types of rock in his frescoes and possibly even the exact location he depicted. In fact, it would be discoveries in the pink Scaglia Rossa limestone--depicted in Giotto's frescoes as pink buildings and used to construct the Basilica of St. Francis at Assisi--which would revolutionize the history of geology.

**1 Introduction**

For a number of years, an international group of scientists and artists have been exploring the many connections between Earth Science and Art. As a geologist, I have long appreciated an artists' ability to capture the beauty of the Earth in a painting, poem or sculpture. While I can analyze the rock formations and marvel at the mystery of the Earth's topography, it takes an artist to move me to tears by presenting their unique view of the

landscape. In doing so, an artist presents the Earth using the visual—a tool so powerful
it can move the most naïve viewer to experience the divine nature of the Earth.
My fascination with St. Francis developed because of his revolutionary ideas regarding
ecology, but I soon learned that without the illustration of his ideas in frescoes by Giotto,
he might not have had such a powerful and lasting impact. The analysis of Giotto's
frescoes from a geological standpoint was a challenge, as St. Francis' monasteries were
located in central Italy, a region that has some of the most complex and still unexplained
geology in the world. Yet, it was precisely this geology that made Giotto's frescoes full of
geomorphological wonder. He integrated his figures into this dramatically disordered
landscape, forming a compelling composition for any viewer. The scenes portrayed in the
frescoes have survived 700 years, allowing modern geologists the opportunity to study
and compare them to formations visible today. In fact, discoveries in the pink Scaglia
Rossa limestone ended up being the key to solving some of the most perplexing questions
in historical geology.
In order to appreciate the revolutionary ideas of St. Francis and Giotto, a background on
the medieval ideas pertaining to nature will be discussed. A select number of frescoes
will be analyzed geologically, with attendant historical information explaining the scene.
Geologic maps which can be used as references for the cited locations are also included.
**1.1 The Idea of Nature in the Middle Ages**
In the Middle Ages, the practice of linking natural phenomena to divine or satanic forces
was the norm (Artz,2014).  Since nature's behavior could not be predicted or controlled,
medieval man lived in a constant state of awareness of its capriciousness. In order to
alleviate stress, a method of spiritual interpretation called "anagoge" was devised by
medieval theologians, notably Hugh of St. Victor (c.1096-1141), to explain natural
occurrences. This meant that one had to search for the of meaning God's messages in
nature through the complex and oftentimes arbitrary symbolism He chose to use
(Cadden,1995). It was thought that by deciphering and diffusing malefic symbols one
could avoid disaster or, in the case of auspicious portents, obtain a fortuitous outcome.
The search for meaning in nature was much more important than the search for "how
nature works" as mechanisms were not valued (Chenu,1983). After all, God was in charge
of everything and what he was doing "behind the scenes" didn't matter. As a result, men
tried to become more empathetic and more closely aligned to nature to understand God.
**1.2 St. Francis**
In the early 1200's, a young man from Assisi named Giovanni di Pietro di Bernardone,
but known to us as Francis, gained a following for his revolutionary ideas pertaining to a
sympathetic view of nature. Francis lived in Umbria, a region of Italy which is green, fertile
and infused with a palpable spirituality. He was born into a well-to-do family of cloth
merchants. As a young man, he renounced his own material wealth, even taking the
position that the Church do the same. He walked to towns and villages, espousing a
simple way of life and encouraging a reverential attitude toward the natural world, for he

believed that nature was the mirror of God. He called all creatures his "brothers and sisters" and preached that people had a duty to protect and enjoy nature as the stewards of God's creation (French,1996). He constructed a series of monasteries (Fusarelli,1999) which were situated in forests or snuggled up against the sides of mountains (Fig. 1). His own cell and bed were carved out of rock. Francis was also a poet and an outstanding innovator in the history of Italian literature. In his *Canticle of the Sun,* believed to be the first work written in the Italian language, he praised God for creating "Brother Sun" and "Sister Moon".

Soon he attracted a group of followers which were organized into many religious orders for both men and women. He became so influential that the Pope had to acknowledge him and allow his orders (Order of Friars Minor, the women's Order of Saint Clare, the Third Order of Saint Francis and the Custody of the Holy Land) to be officially recognized by the Church.

Francis was so venerated that Pope Gregory IX canonized him in 1228, only two years after his death and ordered a basilica be constructed in Assisi. It was built with indigenous pink Scaglia Rossa limestone (Fig. 2) and completed in 1253. Astonishingly, this Scaglia Rossa limestone, depicted in Giotto's frescoes, held the key which would revolutionize the history of geology.

**1.3 Giotto**

The Pope ordered scenes of the life of St. Francis to decorate the interior of the basilica. It was covered with frescoes painted by several generations of Italian artists. Among the many famous names who worked in the Assisi basilica were Cimabue (1240-1302), Duccio (c.1255-1319) and Giotto (1267-1337). They sought to honor St. Francis by portraying his life in a series of frescoes which not only served a proselytizing function but changed the history of art. This one building became the most fruitful single training school and meeting place in the history of Western art (Moleta,1983).

Giotto's works were so revolutionary that today he is considered the founder of Renaissance art (Moleta,1983). But the seeds of this dramatic stylistic transformation were planted by Cimabue, who worked at Assisi during the Pontificate (1288-1292) of Nicolas IV, the first Franciscan Pope. Cimabue broke from the rigidness of Byzantine art where figures were rendered flat and one dimensional (Dall'Asta,2012). Instead, he made them more realistic by using proportions and shading for volume.

Giotto followed Cimabue's lead in changing the flat figures of the Byzantine art form, conducive to contemplation, to a more relaxed style by showing man in a natural setting (Dall'Asta,2012). He observed the landscape and sought to portray its beauty and order. He used painting techniques to depict figures more solidly, so that they would appear three dimensional and thus, created the illusion that they were moving through a landscape within the picture. Giotto integrated sacred images into the Earthly landscape, separating them definitively from their abstract representation in Byzantine art. By portraying daily life, the realm of the sacred appears to be diminished, but that of the

profane acquires a grand dignity and seriousness, which became Giotto's distinctive
characteristic.
Additionally, both Duccio, (a contemporary of Giotto's), and Giotto were among the first
to add a depth perspective to their paintings this was revolutionary, as the rules of
perspective had been lost in the Dark Ages (Blatt,1984). Duccio used modeling (playing
with light and dark colors) to reveal the physique beneath the clothing's heavy drapery.
Hands, faces and feet became more rounded and three dimensional, giving his figures
vitality. Giotto used various techniques, such as presenting his buildings obliquely to take
up more space in depth, and presenting his figures with volume, scale and perspective to
evoke realism (Egerton, 1993). The two artists also placed their figures within natural
settings paying special attention to plants, trees, animals and making detailed rock
formations an integral part of the scene. As a result, their figures interacted with one
another, creating a sense of fluidity and movement in a realistic landscape, which was
astounding to the medieval viewer.
The inclusion of natural elements in visual art as a method of disseminating the new ideas
of St. Francis was very effective, as the general populace was illiterate. Seeing frescoes
reflecting their everyday lives in familiar landscapes changed their way of thinking
(Panofsky,1997). The trees, plants, animals and rocky land forms which had never been
valued suddenly became part of the incredible universe that God had created (Crombie,
1990). The Earth, and all things living upon its glorious terrain were gifts from the Creator
to be used, enjoyed and respected. Duccio and Giotto used nature as a symbol, as the
stage on which the mystery of life, both spiritual and temporal was played out.  A lake, a
mountain range, a valley, depicted in a realistic manner would make the viewer identify
with the painting.  With enough detail, the viewer could find any number of things that
reminded him of his village, farm or pasture. His journey of discovery would make him not
only feel a kinship with the work, but also a sense of conviction that the work was real.
The authenticity of the landscape contributed to disseminating the gospel by convincing
the viewer that the sacred message contained therein was valid, be it Christ's birth or
crucifixion or an episode in the life of a saint. And so, the depiction of these sacred scenes
acted as a catalyst for changes in Western piety as well as Western art (Moleta,1983).
Considering how venerated Francis was, one would think that Giotto would portray him
as a sacred figure surrounded by elegant surroundings, or embellished churches worthy
of exalted adoration.  But Giotto could not do this because of the way Francis lived.  The
stories and legends pertaining to his life and the humble places which he inhabited
required Giotto to place him in nature to depict his life accurately.  So, we see in Giotto's
works the visual story of the ecological movement started by Francis.
In looking at the geology of central Italy, which Giotto portrayed in the scenes of the life
of St. Francis, we can see that he had a keen eye for geologic formations and took the
time to portray them realistically. These regions have some of the most complex geology
in the world, for the Apennine Mountains are not only seismically active but are being
contorted by forces of both compression and extension.  There are faults, upthrusts and
nappes in the area, all of which displace strata, making it difficult for geologists to interpret
the regional geology. Yet in Giotto's works we see that he found these unique geologic
formations perfect for providing visual interest, yet accurate enough for geologists to
identify the rock types and speculate as to the exact location represented in the scenes.
To better understand the placement of the selected Franciscan monasteries and Giotto's
portrayal of the landscape refer to the geologic maps (Figs. 4, 5,6).

## 1.4 Geology

The Apennines, which form the backbone of the peninsula are some 1,200 kilometers in
length.  They were formed some 20 Mya by processes which have not been completely
understood even today, in that they were formed both by compression and extension. In
the east, anticlinal ridges formed from thrust faults create a series of progressive folds,
one after another, while in the west, fault block mountains are created by normal faults
which slice out of the deep crust (Pizzorusso, 2013).
The folds in the Apennines are caused by thrust faulting (Fig.3) where the thrust cuts
upward at about a 30-degree angle called a ramp. This wedge of thrust-up rock is forced
into the shape of an anticline and thus the Apennine ridges are ramp anticlines.  This
range of large anticlinal folds includes Monte Subasio the location of Assisi (Lena et al.,
2014).  They extend from southwest to northeast.  This orderly sequence occurs when
material deposited while building the anticline becomes too large for continued
displacement and the thrust fault slices a new ramp along weaker strata.  The process
continues, with a set of parallel folds visible at the surface forming "propagating fold-thrust
belts" which slice out of the deep crust (Alvarez, 2008).
In the Apennines, as the migrating compressional front moves northeastward (the
anticlines get younger from Gubbio to the Adriatic Sea), there is an extensional front
following behind (about 100 km to the rear).  When the extensional front arrives, the fold
is cut apart by normal faults and subsides as the underlying strata is stretched thin.  The
most likely explanation for this dynamic is that the lower part of the continental crust peels
off and sinks.  This hypothetical process is called "delamination" (Alvarez, 2008).
As noted previously (Fig. 2) the Scaglia Rossa limestone was used in the construction of
the Basilica of St. Francis in Assisi. But amazingly, this limestone would provide the
material for researchers to more accurately date the movement of continents as well as
further their knowledge about the rate at which geologic change occurred.
The Scaglia Rossa is a pelagic sedimentary rock composed of 1-20% foraminifera and
5% clay in a coccolith matrix, deposited from the late Cretaceous to middle Eocene. Its
color ranges from brick red to pink and also grey, yellow and mixed white and pink. The
red and pink colors are due to the oxidation of the iron minerals limonite and hematite.
Because its deposition was undisturbed by erosional gaps, and it was filled with plankton
suited for dating and correlation over long distances, it carried a record of magnetic field
reversals which allowed researchers to obtain data on 100 Myr of geomagnetic polarity
stratigraphy (from c. 137- c. 23 Ma). This data was then used to affix dates on
reconstructed maps of the continental positions since the breakup of Pangea (Alvarez,
221  2009).

Some 65 Mya a giant meteorite hit the Earth, sending smoke, dust and a rare element,
iridium, into the atmosphere. The pollution blocked the sun which resulted in widespread
plant and animal death on Earth, including the dinosaurs. Remarkably, high levels of
iridium (c. 455 ppb in a meteorite vs. c.0.3 ppb in the Earth's crust), indicative of a
meteorite strike were found in the Scaglia Rossa strata dating to 65 Mya, the approximate
time of the extinction of the dinosaurs (Alvarez, 2008).
Another unusual landform with a unique stratigraphy found in the area are foreign rocks
of many different ages are called Ligurides.  They were deposited in an ocean to the west
of Italy—one that no longer exists. They are composed of turbidites and pieces of ocean
crust dating from Triassic to Eocene that were displaced when the ocean was squeezed
shut.  They have slid almost the entire way across the Italian Peninsula. As underlying
anticlinal ridges rose, the fragments would slide down the front toward the northeast.
Sometimes they are badly damaged and sometimes they are found in enormous blocks
(Alvarez, 2008).
In the area around Assisi, (Fig.4) Mt. Subasio, is an anticlinal fold of marine limestone
(pink, cream, gray) formed 10-15 Mya which dominates the landscape. The structural
setting is complex as it is cut in half by normal faults. Today, the eastern half of the
anticline stands while the western half dropped down to the valley below the town. The
area is seismically active. The Basilica of St. Francis of Assisi was constructed with Mt.
Subasio's pink Scaglia Rossa limestone (G. Lena et al., 2014).
The Rieti basin (Figs. 5,10,11) is an intramontane depression of the Apennine chain and
home to a number of Franciscan monasteries. It is filled with continental plio-Quarternary
sediments made of conglomerates, sands, silts and travertine deposits that reach a
thickness of 400-500 meters. The origin and evolution of the Rieti basin is related to the
post-collisional extensional tectonics that have strongly affected this section of the
Apennine orogenic belt since the Pliocene. From the middle Pleistocene to Present along
the course of the Velino river numerous travertine thresholds accreted controlled by
alternating erosional and sedimentary phases (Mancini et al., 2009).
The monastery at La Verna sits on Mt. Penna (Fig.6,17), a Miocene calarenite. It is highly
fractured and many caverns and clefts are etched into its surface. Boulders and scree
surround the base of the mountain. It rests on Cretaceous successions belonging to the
eastern Ligurian Units (Sillano Formation, Early Cretaceous) (Brogi, et al., 2010).
As to the lithologic commentary on the art works, here are the types of deposits which
can be seen aboveground in the referenced areas: travertine, conglomerates, sands, silts,
dolomite, limestone, fluvio-lacustrine deposits, turbidites, carbonates, calcareous tufa,
evaporates, anhydrites, dolostones, marls, sandstone, (basement crystalline and volcanic
rocks which cannot be seen, have been left out). The oldest above ground deposits date
to the Triassic (252-201 Mya) (Carrara et al., 2004).
Comments on the strata will be categorized based on color, form and congruity with the
known geologic conditions in the area since the exact lithology in a Giotto work cannot be
determined with certainty.
For scenes depicting events in the region of the Holy Land, the rocks Giotto portrays are
devoid of vegetation reflective of the desert environment, all the while showing bedding
planes, erosional features and other realistic detail.
The following works by Giotto are a small sampling of his extraordinary output. They were
chosen because of his inclusion of geologic formations and natural elements.

## 1.5 Nativity

Francis staged the first living Nativity scene or *presepe* on Christmas in 1223 in a
limestone grotto at his monastery at Greccio (Fig.7). Interestingly, Francis had to obtain
papal permission to use an ox and an ass in the manger scene to avoid the charge of
novelty. Once approved, he invited the local townspeople, along with their animals, to
participate in a recreation of the holy event. He situated the participants, including
livestock, in the grotto and then placed a newborn in a manger cushioned with hay. After,
Francis stepped forward and lead a celebratory mass. The altar was a block of limestone,
still visible today. This brought the message of Jesus' birth down to Earth so that the
lowliest person could identify with the humble manner in which He was born.
If we look at a Byzantine representation of the Nativity (first part of the 14[th] c. Fig.8) we
can see Jesus' birth depicted in a cavern in a landscape complete with rocks, mountains
and trees. The Byzantine style, lacking perspective and scale, portrayed the figures and
landscape elements one-dimensionally, configured in a single plane (Dall'Asta,2012). In
religious art, this effectively created a psychological distance between the sacred events
and the viewer, evoking a reverential experience.
Giotto revolutionized art by taking Byzantine iconography and humanizing it (Fig.9).
Following Francis' lead, the Nativity thus became a natural event. Using elementary
perspective techniques, he was able to compose a sacred scene that appeared similar to
a person's daily life. In this way, the viewer had a direct experience with the miraculous,
allowing him to internalize the supernatural event and ultimately transfigure his human
consciousness into a vessel for the divine (Panofsky,1997).
Giotto also revolutionized the depiction of natural elements by including them as vital to
the composition, and also applying the same techniques-- perspective, shading, etc. on
them as he used on his figures. This rendered the scene realistic and the location was
often identifiable to the locals. In his portrayal of the Nativity he reproduced the geology
of the area surrounding the monastery at Greccio (Fig.10,11) which consists of carbonate
units of the Sabina Sequence (Meso-Cenozoic) (Carrara et al., 2004) (Falcetti et al.,
296 2014).

He depicted a limestone ledge and added a rudimentary wooden roof for shelter. The limestone strata in the background are upthrust as shown by the vertical relief. These blocks, formed by the dynamic movement of the earth, now act as a sheltering backdrop for the manger holding the newborn. Angels also hover overhead to protect, pray and rejoice at the miraculous event. The ox and donkey on the left are farm animals, vital to the sustenance of the people. The sheep, goats and their shepherds were also common to the area. Today, going to the monastery at Greccio, (Fig.10) one can see the limestone cliffs, crevasses as well as the original grotto that inspired St. Francis.

In describing the Nativity, we are told that Mary and Joseph embarked on a journey, the night was cold and starry, there was no room in an inn nor help with the birth. The lowly manger was filled with hay and animals were settling in for the night. Here, we see that Giotto continues the theme of Jesus' birth in a limestone landscape (Fig.12). The upthrust block in the background provides shelter for the newborn set upon an altar-like formation of the bedrock in the foreground. And so, Jesus was born without fanfare as people went about their daily tasks. He did not stop the world, rather He changed its orientation and sensibility. Men continued to eat, talk and work, live and die, yet the birth of Jesus changed the intrinsic purpose of their actions and their lives. Placing Jesus in a manger, the locus where animals were fed, let us know that He would provide us with food as well (his body). The gospel of St. John 1:9 tells us: "there came into the world the true light (external light) which illuminates every man".  Meaning that with the birth of Jesus, divine light appeared on Earth and was the vehicle used to communicate the gift of divine life. In the story of Creation, the contrast between darkness and light was used as a metaphor. Now, in the mystery of the Nativity it returned, and was transfigured into a more intimate form (might be considered internal light) where God enters into the lives of men to create a second definitive creation.  John 8:12 says "I am the light of the world, he who follows me does not walk in darkness but will have the light of his life." Giotto deftly incorporated light into his scenes to illustrate gospel teachings as well as well as miraculous events.

Another revolution in the portrayal of the Nativity was the change, in the 14[th] century, from the use of a cavern, to a "inn" (*kataluma*) as described in the Gospel of Luke. From that, the location was often a "diversorium" which might be an inn, a cabin (*capanna)*, or a hut with a canopy (*tettoia*) which were common in medieval cities (Dall'Asta, 2012). These were public places where people came to rest and talk. These became the new churches, humble and unpretentious, according to the reform principles of the Franciscans who longed to return to a simple evangelization.

In this fresco, we pass from a desert, an isolated locale, to an urban setting (Fig.13). The abandonment of the desert and the grotto has a precise theological justification.  By placing Jesus' birth in a city, not in the wilderness, the mystery of his divine nature would not be hidden from the people.  He is portrayed as being born in a town, near a market, in an open, populated place where his nature can be seen by all.

The baby is often placed in the foreground on the earth, underlining his human character, propped on a bale of hay-- an illusion to the eucharistic bread, or on a sheet--evocative

of the shroud.  In this manner, if the faithful looked down, they would have understood
the humility of the divine birth.  From an etymological standpoint, the word "humble" can
be taken to mean "attached/close to the ground" (in Latin, *humus*).

**1.6 Preaching to the Birds**

In Byzantine art, the background was usually gold, a glorious, expensive color which
invoked a sense of awe of the Divine and, as a result, kept the viewer at a reverential
distance (Dall'Asta, 2012).  As a color, it was flat which did not draw the viewer into the
scene. Giotto's treatment of this event (Fig.14) is very interesting because of his use of a
gold background. The gold finish is textured and shaded and the dark foreground cuts a
horizontal band, imparting depth and three dimensionality. He then places the tree in a
manner in which it is growing out of the picture. St. Francis is preaching to birds who are
walking and flying toward him, seemingly enraptured by his words. Due to the use of
color, shading and perspective, Giotto created a work that had volume and movement.
The tree is swaying in the wind, the birds are flying and walking and the friar behind St.
Francis is in a different plane, giving the whole picture a sense of depth and dynamism.
The warm colors invoke an autumn day with an orange-gold sun illuminating the
background. While the use of earth tones and touches of dark gray-greens give the work
a cohesiveness, warmth and intimacy. One wants to watch, an experience we have all
had while viewing flocks of birds, yet we want to be still and quiet so as to not disturb
them lest they fly away. In this manner, Giotto works his magic, allowing us to feel the
peace and mystical nature of God's Earth and His creations by presenting them in a
simple setting that is reminiscent of our everyday life.
An incident illustrating Francis' benevolent attitude towards nature is recounted in the
*Fioretti di San Francesco* (The Little Flowers of St. Francis), a collection of legends and
folklore that was compiled after his death. One day, while Francis was traveling with some
companions, they happened upon a place in the road where birds filled the trees. He told
his companions to "wait for me while I go to preach to my sisters the birds."  The birds
surrounded him, intrigued by the power of his voice, and not one of them flew away.

**1.7 The Flight into Egypt**

The Gospel of Matthew 2:13-23 recounts that after the visit of the Magi to the newborn
child, an angel appeared to Joseph in a dream and told him to flee to Egypt with Mary
and Jesus, as King Herod would seek to kill the child.  In this scene, Giotto portrays an
arid landscape (Fig.15). The mountains are sparsely vegetated and the desert through
which they are traveling is inhospitable. It is an arduous journey with the donkey making
its way along a narrow path with a steep precipice in the foreground. Joseph leads the
way with an angel, most likely the one which appeared to him in the dream, guiding and
protecting them on their way. Mary and the child sit upright, with great dignity as they
endure the harsh traveling conditions. Giotto chooses a background of gray and blue to
impart the sensation of a rocky, barren landscape where even the few trees must struggle
to survive. He pays attention to the rock strata and bedding planes so that the formations

would appear close to those we see in nature. The dark blue sky and impending darkness causes a sense of preoccupation for the welfare of the family. Were they traveling by night to avoid detection or avoid the harsh sun? This is a mystery. Where will they rest? There are no buildings or indications they are close to a village or city. So here, Giotto presents a barren, dark, uninviting environment that would have been unfamiliar to the Italians living in the florid Italian countryside. Yet, they would understand the hardship involved for a mother and newborn to undertake this journey on a donkey. The vast unknown terrain, with no water or vegetation to sustain them, leaves the viewer sympathizing with the Holy Family and respecting the sacrifice they made for our ultimate salvation.

**1.8 The Dream of Joachim**

This touching scene shows St. Joachim, husband of St. Ann and father of the Blessed Virgin Mary, in exile in the wilderness (Fig.16). The landscape, colors, and posture of St. Joachim convey a profound sense of despair. St. Joachim and St. Ann had reached advanced ages without having a child.  This was considered an indication of God's wrath. Joachim went to the temple to make a sacrifice, which was rejected, and he was then expelled by the rabbis. He went into exile in the mountains leaving behind his wife, Ann. As we can see, Giotto places him in a hunched-over position with his head resting on his knees. He is desperate, inconsolable. He sits directly on the ground, is he so weak or defeated that he no longer can or will get up? The landscape is stark and a dark mountain with no vegetation rises menacingly in the background. The carbonate rocks in the fore and middle ground are lighter, reflecting their natural color, but arid, save for a very few trees. The only people in view are the shepherds who frequented the mountains with their flocks. The small cabin is made of blocks of limestone likely mined from the local area. Perhaps it was the "*refugio*" or cabin of the shepherd who used it at night. One of the sheep appears to be entering a grike (solution fissure). Giotto portrays the natural landscape here as barren, a metaphor for the fruitless matrimony of Joachim and Ann. He does a marvelous job depicting the nearly vertical bedding planes of the dark brown formation, perhaps a bedded sandstone, in the distant background. Geologically, the beds were originally laid down flat, and with subsequent deformation and movement they were thrust upward into their nearly vertical configuration. Giotto depicts the carbonates in the foreground as they appear in nature, blocky, with cracks and crevasses and, where it has been eroded by wind or rain, has softer edges.  Giotto creates a masterful geologic environment, paying careful attention to the physical characteristics of the different types of rock. What hope can there be in such an environment where there is no sign of fertility, no lush green plants, no water—nothing. There is something however, the angel. It is bringing word to Joachim that Ann is with child and she will be blessed. Joachim's world will change with this message and our world will be changed as well.

**1.9 St. Francis Receiving the Stigmata**

The grotto of the monastery at La Verna was the place at which St. Francis received the stigmata of Christ in 1224. La Verna, where today, pilgrims still visit to pray and meditate

is located on Mt. Penna (Fig.6) in the Apennine ridge connecting Casentino and Valtiberina. In ancient times, people couldn't explain how this mount, a mass of limestone, came to be, so the legend was born that it (Mount Alvernia in Latin), geographically known as Mt. Penna was created by a strong earthquake occurring when Jesus died on the cross. Its geological origins are so complex that even today, scholars are still trying to decipher it.  However, research (Brogi & Fabbrini, 2010) indicates that Mt. Penna (Fig.17) is composed of Miocene calcarenite resting Cretaceous successions belonging to the eastern Ligurian Units (Sillano Formation, Early Cretaceous)

In this image of solitary mystical experience (Fig. 18), Giotto portrays Francis on a block of limestone which has been weathered and uplifted as seen by its nearly vertical relief. A cleft in the side of the cliff, common to calcareous deposits, has opened. Giotto uses this rock, which has been sliced open, to imitate the wounds in St. Francis' hands and feet. The church in the foreground is made of the gray limestone found in the area and commonly used for construction. To the left of the church grikes (solution fissures) and clints (limestone separated from adjacent sections by solution fissures) are starting to form. Behind the kneeling figure is the cave where, in one account, he struggled nightly with demons.  Above the cave perches the falcon which woke him for his vigils, and whose hovering flutter was an omen of the heights of contemplation to which Francis would soar. Flora and fauna are sparse and the sky is a deep gray black forcing us to pay attention to the miracle that is playing out on this mountainside. The Franciscans used this location and divine occurrence to demonstrate that mountains were vital in the sacred ritual, thus promulgating the idea that they would provide a nearness to God and a source of divine inspiration (Schama,1995).

An excerpt from the anthology "*Fioretti di San Francesco"* (The Little Flowers of St. Francis, anonymous medieval manuscript), describes this miracle:

"considering the form of the mountain and marveling at the
exceeding great clefts and caverns in the mighty rocks, he betook
himself to prayer and it was revealed to him that those clefts…
had been miraculously made at the hour of the Passion of Christ
when, according to the gospel, the rocks were rent asunder."

**1.10 St. Francis Gives His Mantle to a Poor Man**

In this scene, Francis demonstrates his commitment to refuting worldly goods by giving his mantle to a poor man (Fig.19). He has abandoned his fine clothing and is now dressed in the simple sackcloth emblematic of the congregation of friars. This is an unwitnessed and spontaneous act which takes place in a rural setting. While art historians claim the town on the hill is Assisi, this would not be accurate as Assisi sits on the western edge of Monte Subasio, anticlinal fold formed 10-15 Mya above the thrust ramps and cut in half by normal faults as the extensional front passed through in the last few million years.  And so, the Monte Subasio we see today is a "half anticline" with the eastern half still standing and western half dropped down to the valley to the west of the town (Alvarez,2008). The actual site may be another location (there are many) or a montage used for dramatic

effect as Francis is placed at the midpoint, between two hills, one with a town and the other with a monastery. He leaves one behind and moves unknowingly toward the other. Giotto uses perspective and scale to depict the town realistically in the distance, complete with the walls which surround it. Remnants of medieval walls such as these, constructed with local material, often limestone, can still be seen today. The towns were historically located on high ground for security. The finely detailed terrain is evocative of the countryside one can see today in central Italy. The rock formations are most likely limestone due to the color, blocky form, faults, grikes and clints. Enormous sections of strata were overturned and displaced as a result of thrust-block mountain building and continuous seismic activity in the region since Roman times (Guidoboni & Ferrari, 2000). The gorges and crevices still visible in many areas today are for the most part unnamed and are best seen untouched in the many national parks, but one, the Bottaccione Gorge near Gubbio, is a mecca for geologists looking at the famous K/T boundary in the Scaglia Rossa limestone.

It is said that Francis walked from one village to another, where he would preach. Giotto places him on a solitary path out of town. In this way, out of sight of anyone, he practiced his charity—anonymously and in the midst of nature. The colors Giotto uses are characteristic of limestone, ranging from milky white to ivory to light gray and pink. The towns would have been constructed with blocks of local calcareous rock so the delicate pastels which characterize the buildings and walls are the actual color of the indigenous rock. In fact, many of the buildings in Giotto's frescoes are pink. The trees hang precariously on the slopes as they endeavor to insert their roots in crevasses and cracks. The misty blue sky is common to the area, where frequent rainfall and clouds add to the mystique of the atmosphere. A scene like this would resonate with any viewer as they would understand the landscape and could recognize the local cities with their houses, churches and towers. They could see familiar mountain paths and remember their own difficult journeys, be them psychological, spiritual or corporeal. And so, through Francis' example, and ultimately through their own actions, seen or unseen, they could become saints as well.

**1.11 The Legend of St. Francis: Miracle of the Spring**

St. Francis, retiring to pray in the wilderness during high summer became ill and was forced to go by donkey (Fig.20). When the farmer who owned the animal begged for water, Francis took pity on him and, after praying, struck a rock and water came bubbling out of the earth. Here, Giotto portrays the landscape in the foreground and middle of the picture with wave-like patterns formed by the erosion of stratified limestone, such as the Maiolica limestone found in many parts of Umbria (Galdenzi,2013). In the background, large blocks have been displaced and turned upright when thrust faults sliced up through the earth. The textures bedding planes and erosional patterns are realistic. In the foreground St. Francis prays on what appears to be and inclined calcareous sinter terrace. In the foreground we see a crevasse which was formed during the ongoing seismic or thrust-block mountain building activity in the area. In the background we see a

dark area between the two rock formations which may be a fault. Limestone is porous
and often springs will gush forth from the interior of the earth. The ground has been
fractured and deformed and many faults have caused displacement of strata as well as
fissures and crevasses. Interestingly, the Italian Secretary of Transportation, Riccardo
Nencini, advanced an idea that the actual location of this spring is the cascade of the
Rovigo torrent in Firenzuola. While this is not supported by geologic data, it is a tribute to
the power of Giotto's imagery, in that, after 700 years, people are attempting to identify
the landscapes he depicted.

## 1.12 The Enduring Legacy of St. Francis and Giotto

The frescoes, altar panels and paintings reflecting the new naturalistic style also provided
visual accompaniment to the popular preaching approach practiced by St. Francis-- not
in Latin, but in the spoken language (Umbrian form of Italian).  Together, the visual and
the audible messages centered on the mystery of the Incarnation and on the need for
repentance.  In fact, the power of the visual representation of nature was much more
powerful than the written word, as most people were illiterate and texts available for study
were for the most part, ancient or ecclesiastical. Aristotle, Pliny and others formed the
basis of natural philosophy and their ideas had not been altered or challenged in 1,500
years (Grant, 2010). With the arrival of St. Francis and Giotto however, a shift in thinking
resulted in massive changes in many disciplines, and nature was one of them (Schama,
1995). When texts on nature started to be published in the Renaissance, the ideas they
set forth were very late in arriving, for the ecological and natural history ideas of St.
Francis as represented artistically by Giotto had already been absorbed into the psyche
of the common man for over 200 years.
Some 750 years after the saint's death, on 29 November 1979, Pope John Paul II
declared Saint Francis the Patron Saint of Ecology. Successive Popes continued to use
St. Francis as a model in their public comments over the years:
"…not to behave like dissident predators where nature is concerned, but to assume
responsibility for it, taking all care so that everything stays healthy and integrated, so as
to offer a welcoming and friendly environment even to those who succeed us."
"As a friend of the poor who was loved by God's creatures, Saint Francis invited all of
creation – animals, plants, natural forces, even Brother Sun and Sister Moon – to give
honor and praise to the Lord. The poor man of Assisi gives us striking witness that when
we are at peace with God we are better able to devote ourselves to building up that peace
with all creation which is inseparable from peace among all peoples."
"It is my hope that the inspiration of Saint Francis will help us to keep ever alive a sense
of 'fraternity' with all those good and beautiful things which Almighty God has created."
"St. Francis teaches us that, the world of God and the world of nature are one."

## 1.13 Conclusion

St. Francis and Giotto, two revolutionary figures who never knew each other, were linked by history and art. Unbeknownst to them, their legacy would ultimately change Western piety, art and natural history. Much of today's ecological movement has embraced the tenets espoused by St. Francis. Giotto not only immortalized Francis' idea of the sacredness of nature by carefully placing and configuring geological elements realistically in his frescoes, he provided a lasting visual record, which allows modern researchers a basis for further study. Not only can they identify the landforms of central Italy, one of the most complicated areas in the world, they know that Giotto's pastel colored buildings were not flights of fancy but duplicated the colors of the indigenous pink, grey and ivory limestone actually used in medieval construction. Amazingly, the use of the pink Scaglia Rossa limestone to build the Basilica of St. Francis would end up being the key to unlocking many mysteries in the history of geology. So while researchers continue to study the outcrops and mountains on which St. Francis build his monasteries, they will also learn that these outcrops were, and still are, miraculously sacred sites. And so, central Italy seems to be a geologist's paradise, where one can participate in the realm of art and religion by looking at Giotto's frescoes and unlock the Earth's mysteries while walking in the footsteps of St. Francis.

## Definitions

Anticline- In structural geology, an anticline is a type of fold that is an arch-like shape and has its oldest beds at its core.

Calcarenite- A type of limestone that is composed predominantly, more than 50 percent, of detrital (transported) sand-size (0.0625 to 2 mm in diameter), carbonate grains. The grains consist of sand-size grains of either corals, shells, pellets, fragments of older limestones and dolomites, other carbonate grains, or some combination of these. Calcarenite is the carbonate equivalent of a sandstone.

Coccoliths- Are individual plates of calcium carbonate formed by coccolithophores (single-celled algae such as Emiliania huxleyi) which are arranged around them in a coccosphere.

Conglomerate-Is a coarse-grained clastic sedimentary rock that is composed of a substantial fraction of rounded to subangular gravel-sized clasts, e.g., granules, pebbles, cobbles and boulders, larger than 2 mm. in diameter.

Fault- Is a planar fracture or discontinuity in a volume of rock across which there has been significant displacement as a result of rock-mass movement.

Fault blocks- Are very large blocks of rock, sometimes hundreds of kilometers in extent, created by tectonic and localized stresses in the Earth's crust.

Foraminifera- Informally called "forams", are members of a phylum or class of amoeboid protists characterized by streaming granular ectoplasm for catching food and other uses; and commonly an external shell (called a "test") of diverse forms and materials.

Marl or marlstone- Is a calcium carbonate or lime-rich mud or mudstone which contains variable amounts of clays and silt. The dominant carbonate mineral in most marls is calcite, but other carbonate minerals such as aragonite, dolomite, and siderite may be present.

Turbidite- Is the geologic deposit of a turbidity current, which is a type of sediment gravity flow responsible for distributing vast amounts of clastic sediment into the deep ocean.

Orogeny- Is the primary mechanism by which mountains are built on continents.

Scree- Is a collection of broken rock fragments at the base of crags, mountain cliffs, volcanoes or valley shoulders that has accumulated through periodic rockfall from adjacent cliff faces.

Tectonics- Is the process that controls the structure and properties of the Earth's crust and its evolution through time. In particular, it describes the processes of mountain building.

Thrust fault- Is a break in the Earth's crust, across which older rocks are pushed above younger rocks.

Travertine- Is a form of limestone deposited by mineral springs, especially hot springs.

Source: Wikipedia

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

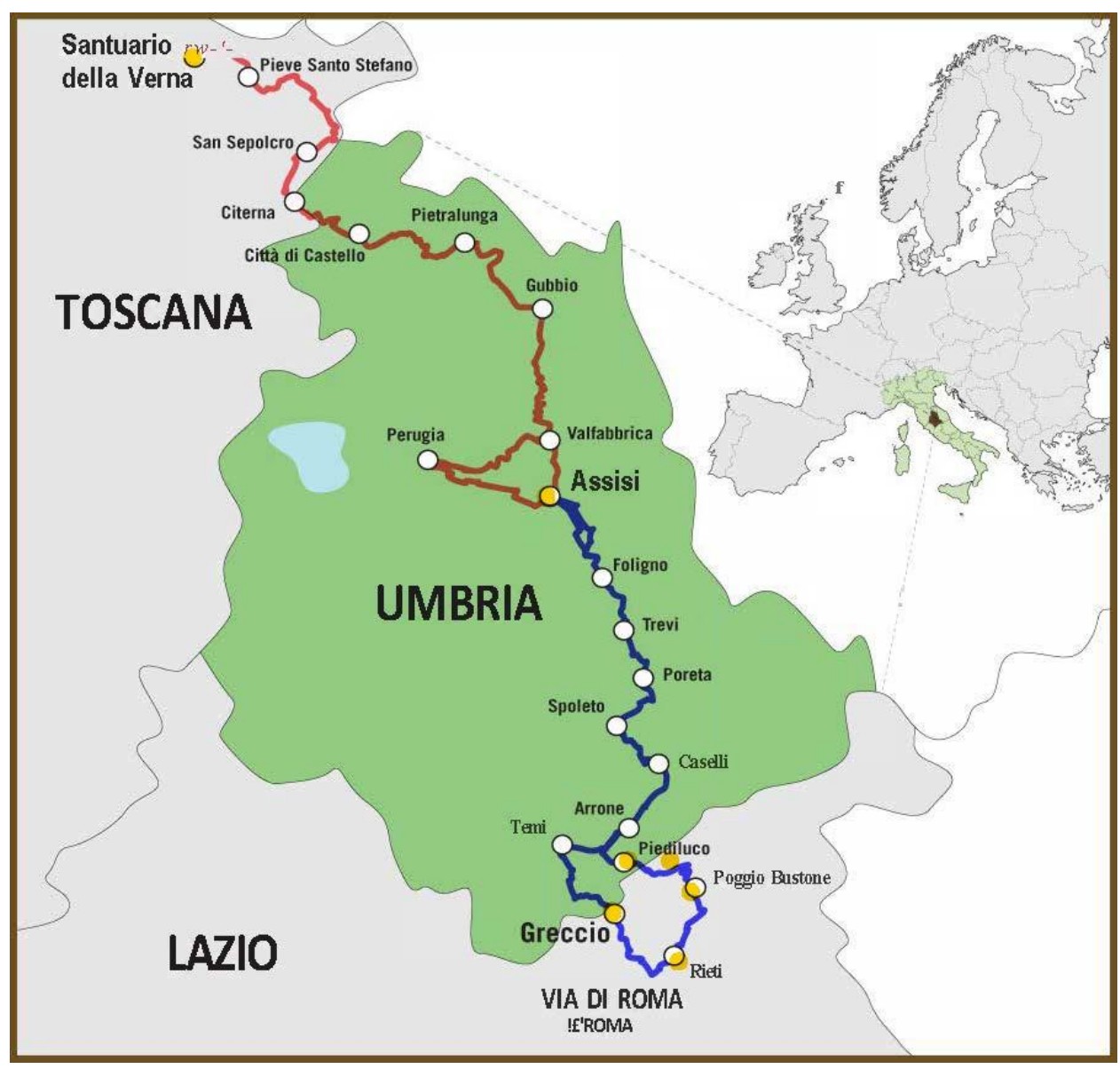

Fig. 1. Map showing selected Franciscan monasteries in Tuscany/Lazio/Umbria (in yellow) and walking paths from one to another.
Note that there are others in Italy which were not chosen as part of this study. Public domain Wiki Commons.


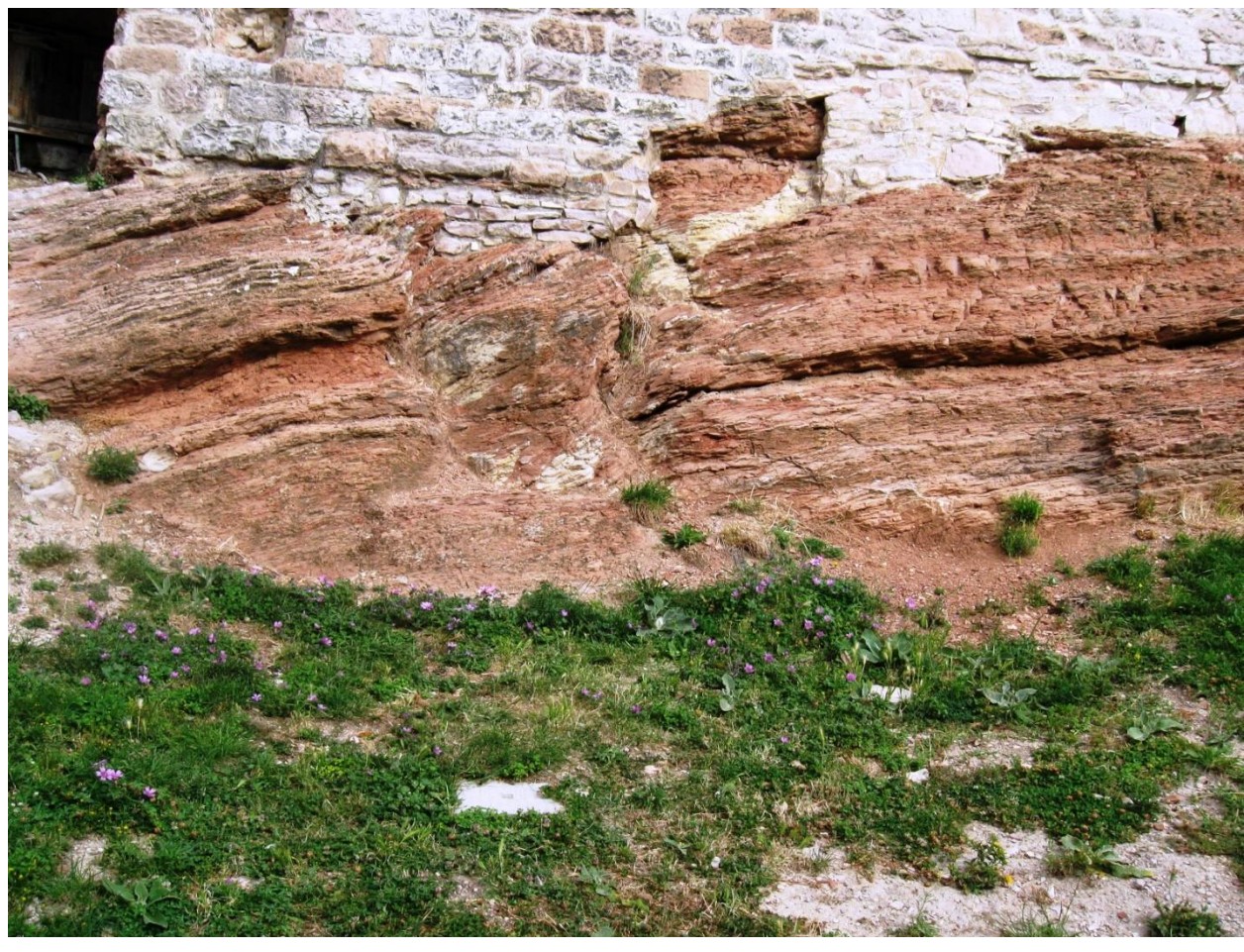


Fig. 2. Pink streaked with white Scaglia Rossa limestone is a pelagic sedimentary rock with forams and clay in a coccolith matix dating
from the Late Cretaceous to middle Eocene. Mined from the Mt. Subasio quarry and used to construct the St. Francis basilica at
Assisi. Public domain Wiki Commons




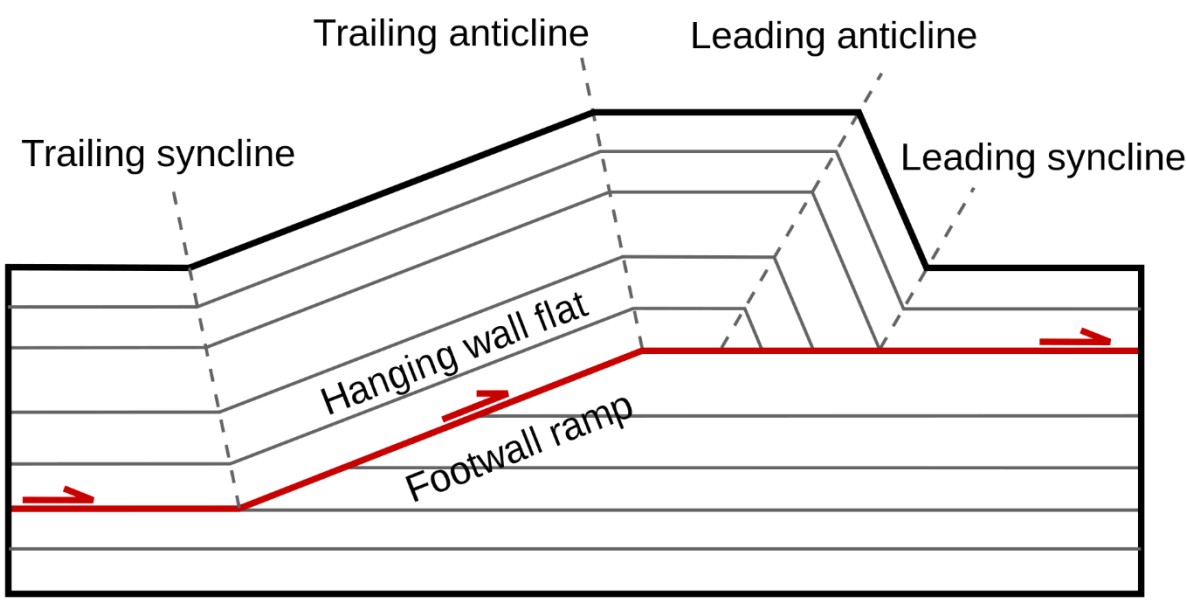

Fault-bend fold


Fig. 3. Diagram of the anticlinal fold mountains formed by compression. Wiki Commons.




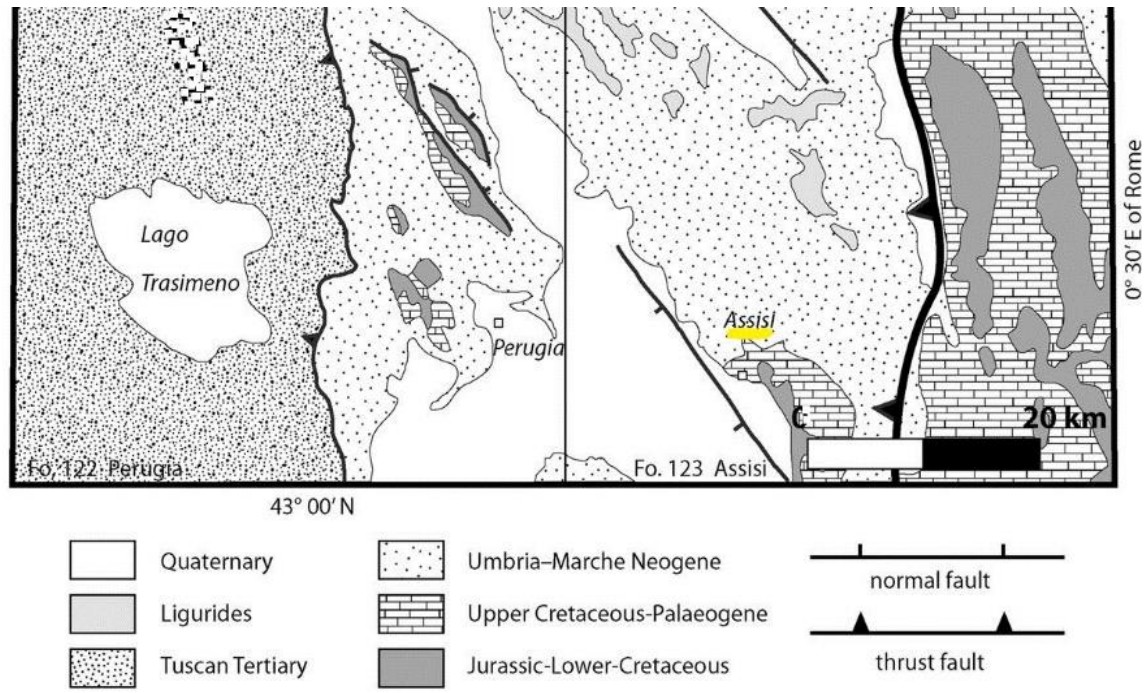


Fig. 4. Geologic map of the area around Assisi. After G. Lena, et al. Geological Society, London 409, November 2014.








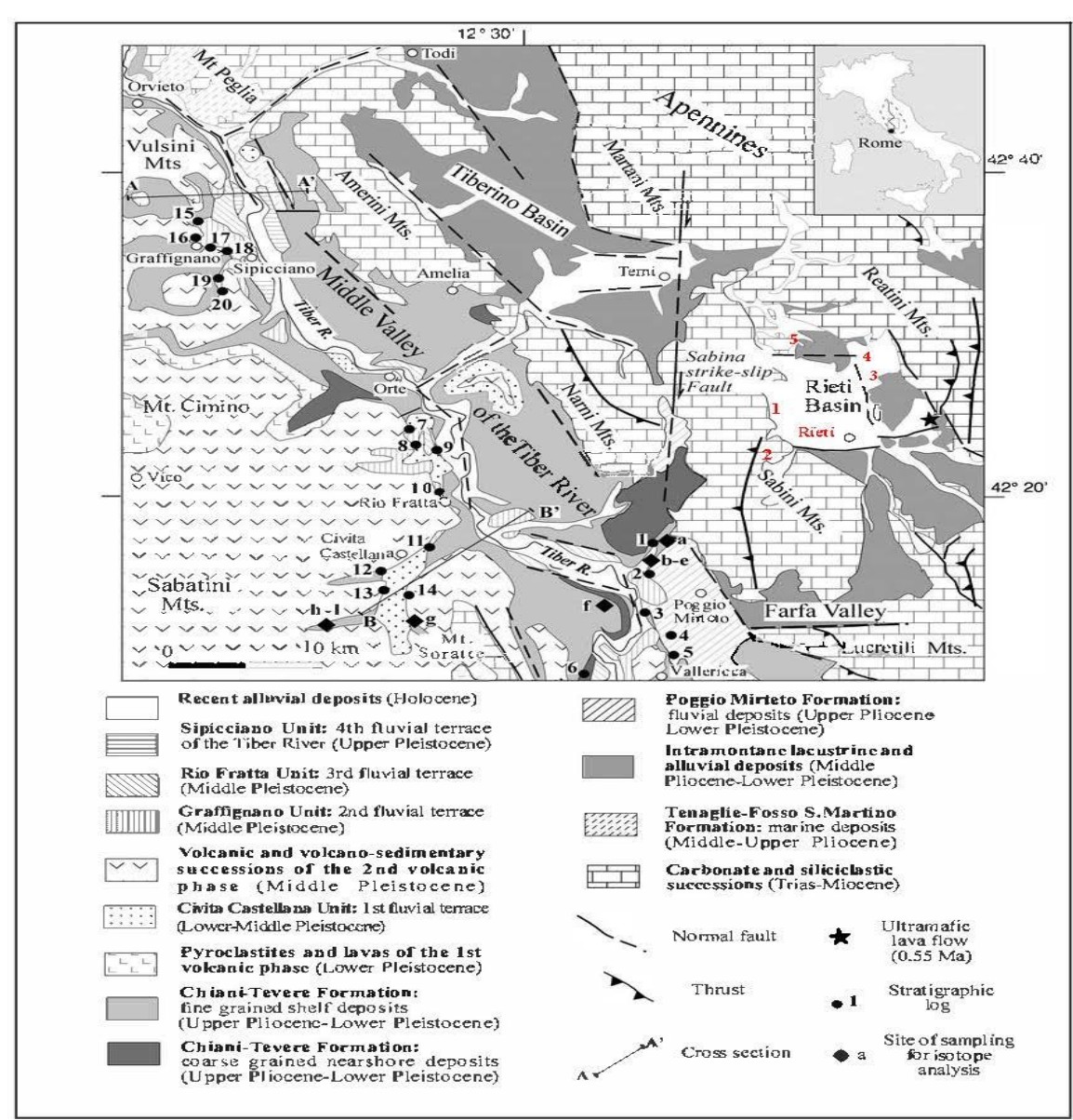



Fig. 5. Franciscan monasteries (in Red) 1) Greccio 2) Santuario di Fonte Colombo 3) Santuario della
Foresta 4) Poggio Bustone 5) Labro. After Mancini, et. al. (2009).




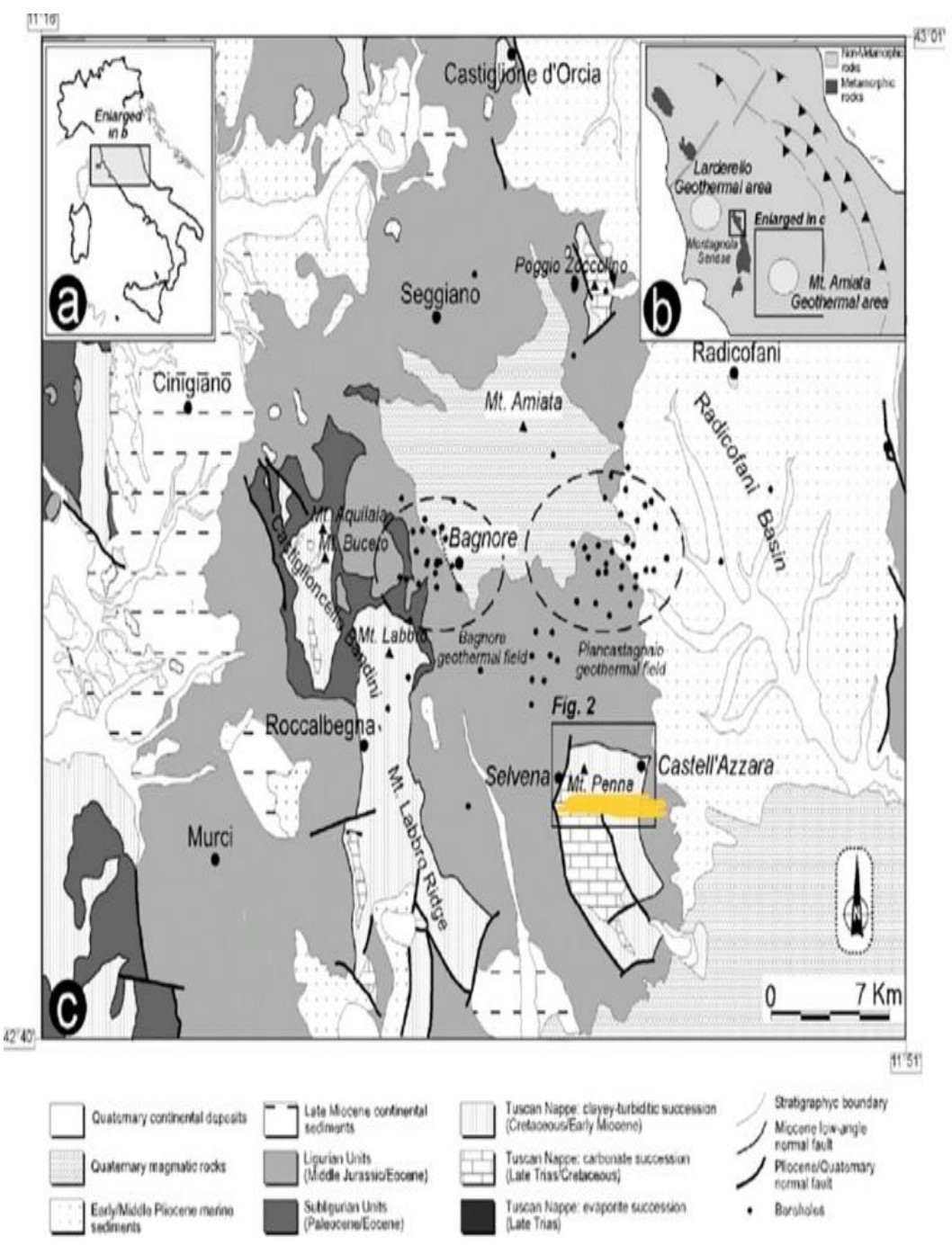



Fig. 6. The monastery at La Verna sits on Mt. Penna, a Miocene calarenite. It is highly fractured and
many caverns and clefts are etched into its surface. Boulders and scree surround the base of the
mountain. It rests on Cretaceous successions belonging to the eastern Ligurian Units (Sillano Formation,
Early Cretaceous). After Brogi, A., et. al. Italian Journal of Geosciences 129(1) 74-90, February 2010.


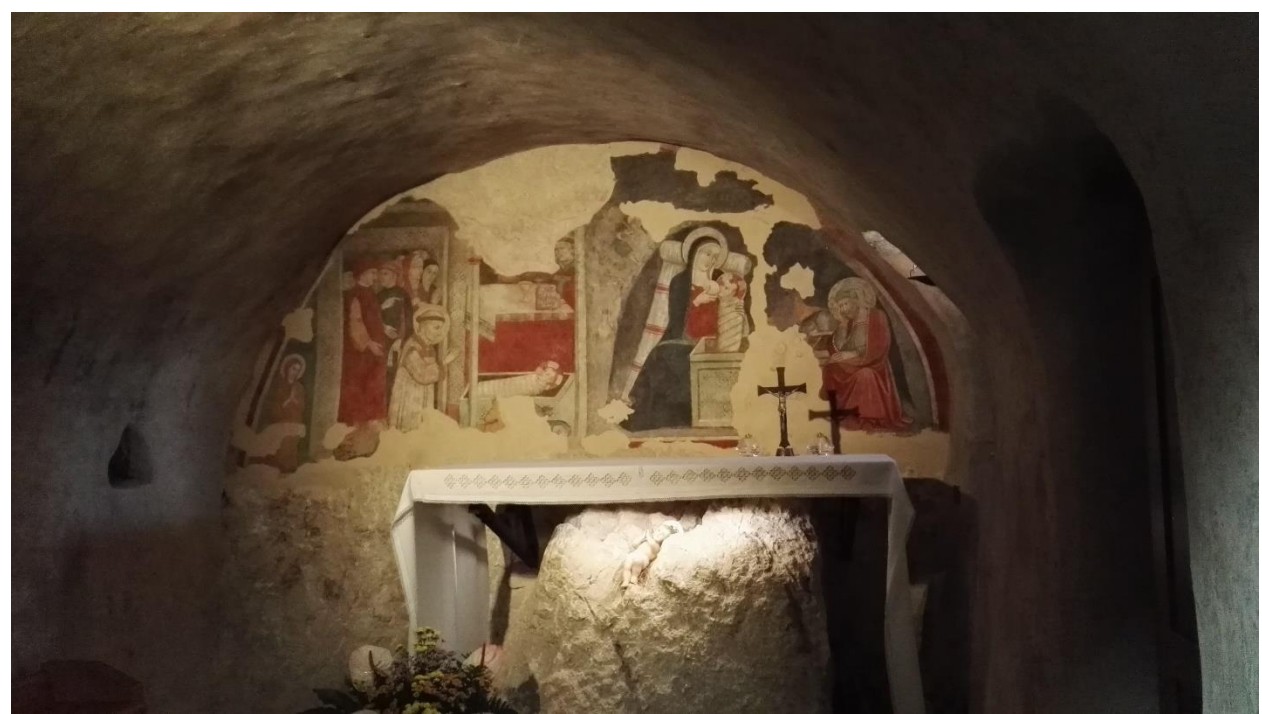


Fig. 7. Limestone grotto at the monastery at Greccio, site of the first Nativity scene organized by Francis on Christmas in 1223. The
limestone outcrop was the original altar before another was placed above it when the Pope visited. The 14[th] century frescoes depict
the original Nativity scene. Photo by Ann C. Pizzorusso.



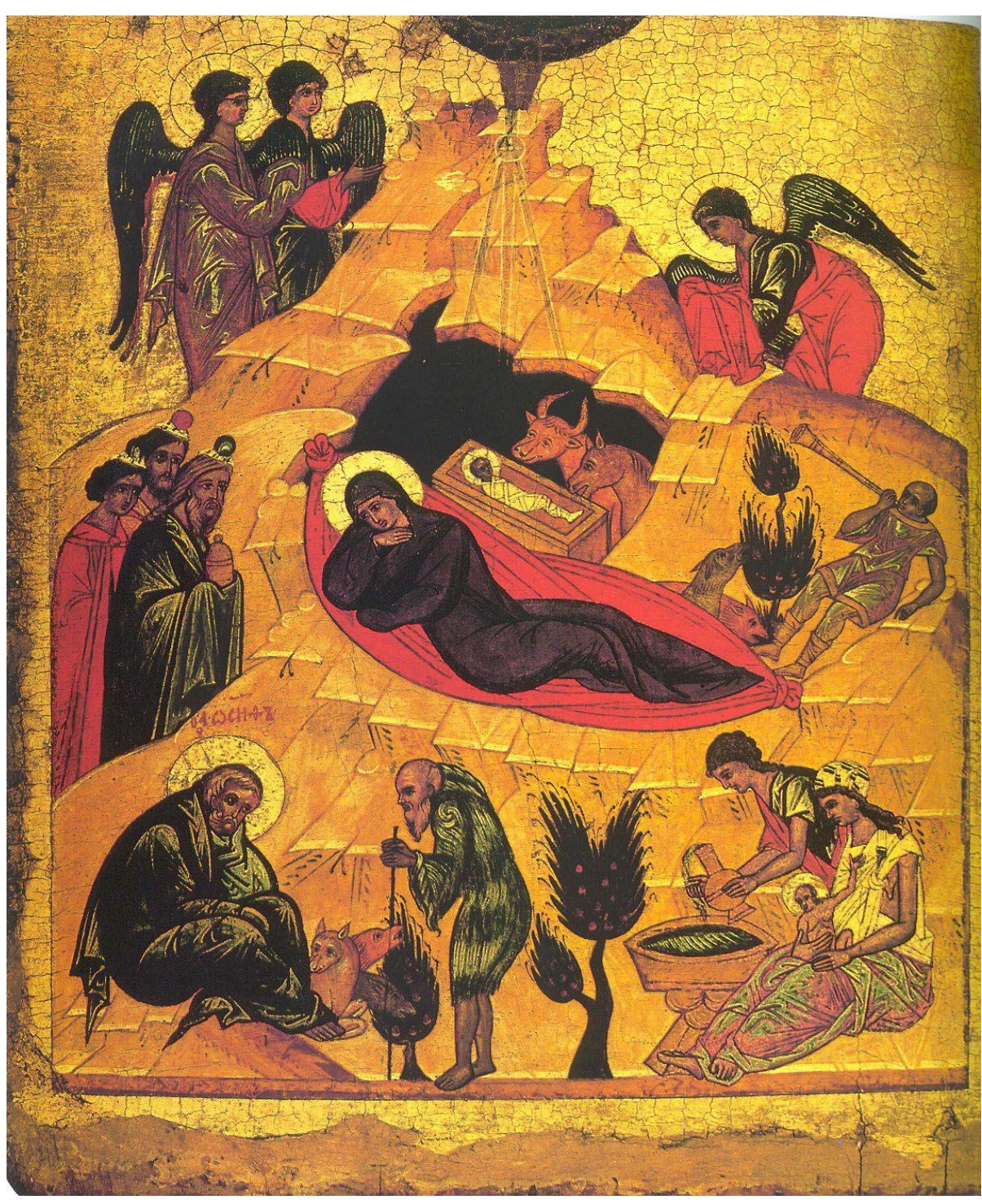



Fig. 8. Nativity. Andrej Rublev. Note how the figures are one dimensional and the entire work lacks perspective. First half of 14th
century. Moscow, Tretjakov Gallery. Public domain Wiki Commons


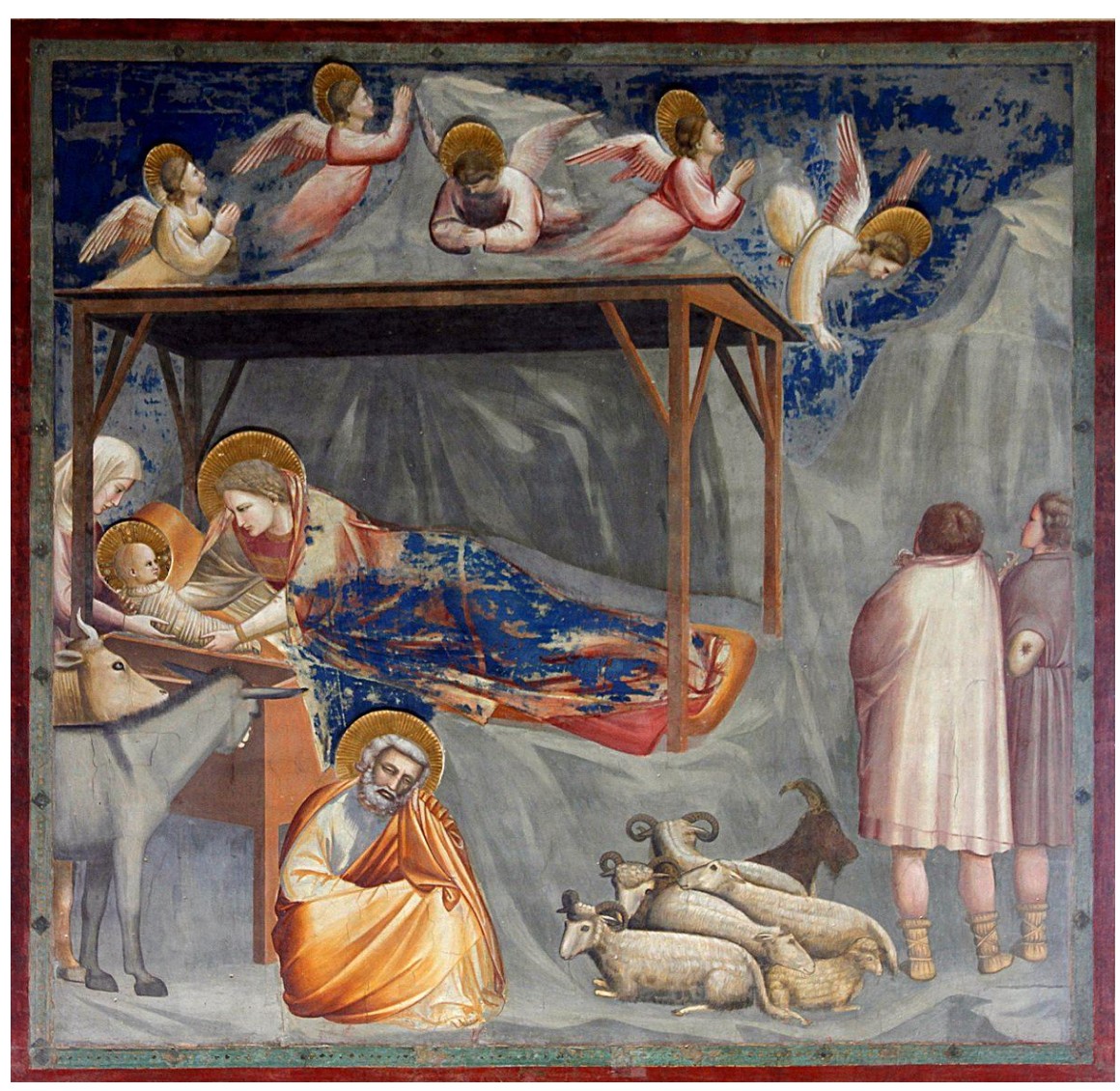



Fig. 9. Nativity. Giotto, c. 1303-c.1306 Scrovegni (Arena) Chapel, Padua, Italy. Public domain Wiki Commons

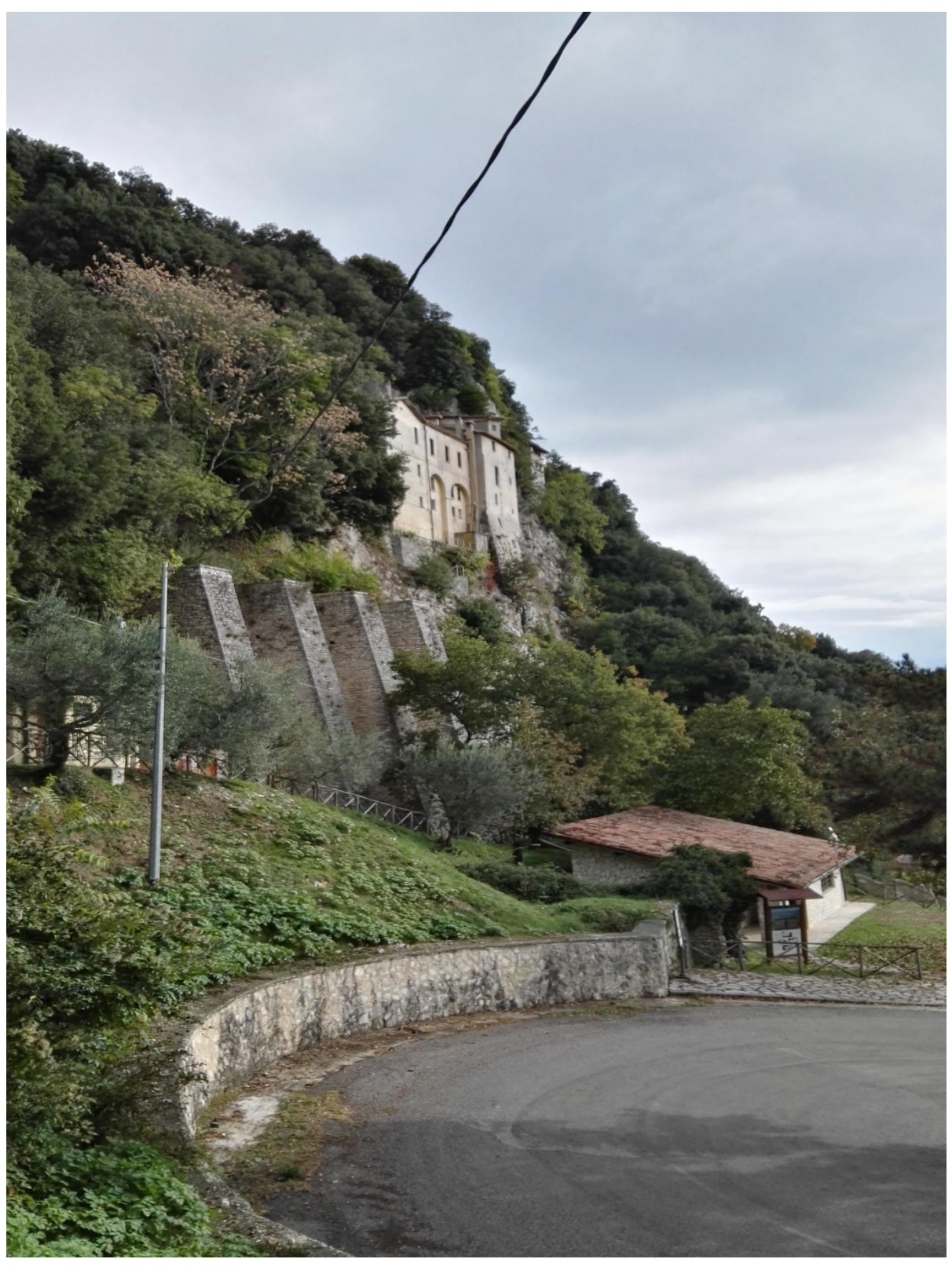


Fig. 10. Greccio, view of the monastery, location of the first living Nativity scene organized by Francis in 1223. The monastery is
located along a thrust fault and is built on carbonate units of the Sabina Sequence (Meso-Cenozoic). See geologic map (Fig.5). Photo
by Ann C. Pizzorusso.

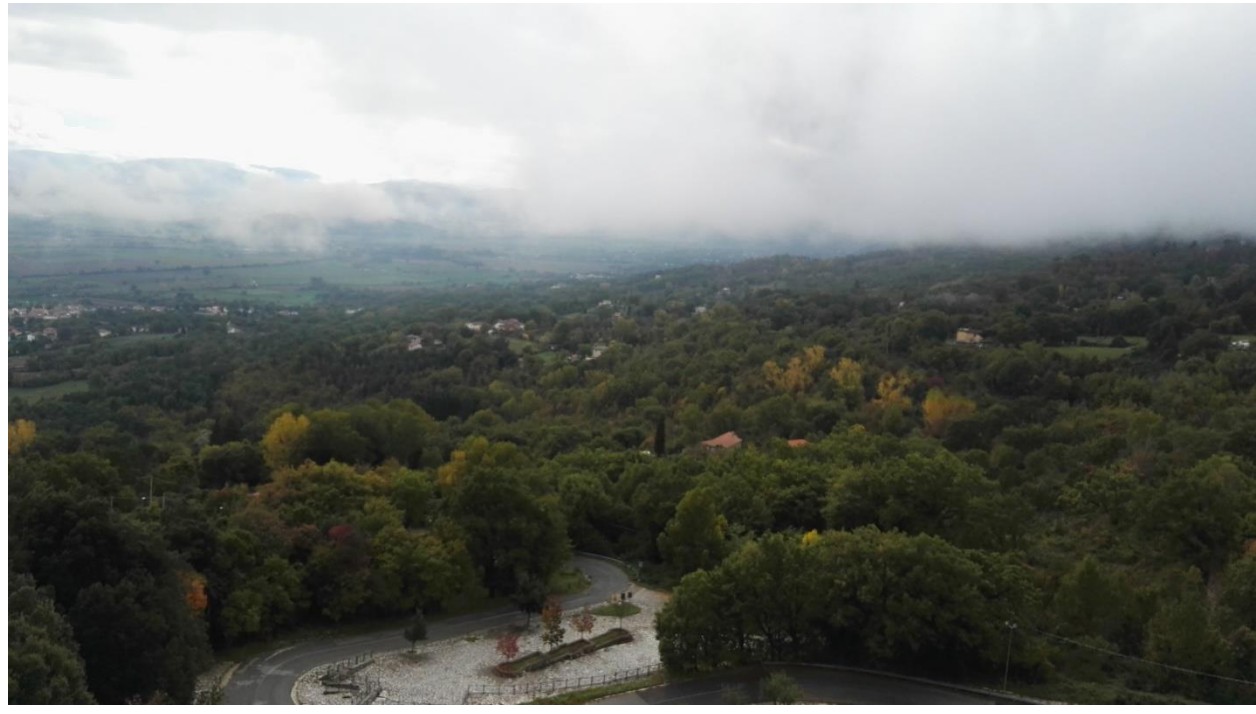


Fig. 11. View from the monastery at Greccio, built on carbonates, looking out at the fluvio-lacustrine and fan deposits. Photo by Ann
C. Pizzorusso.



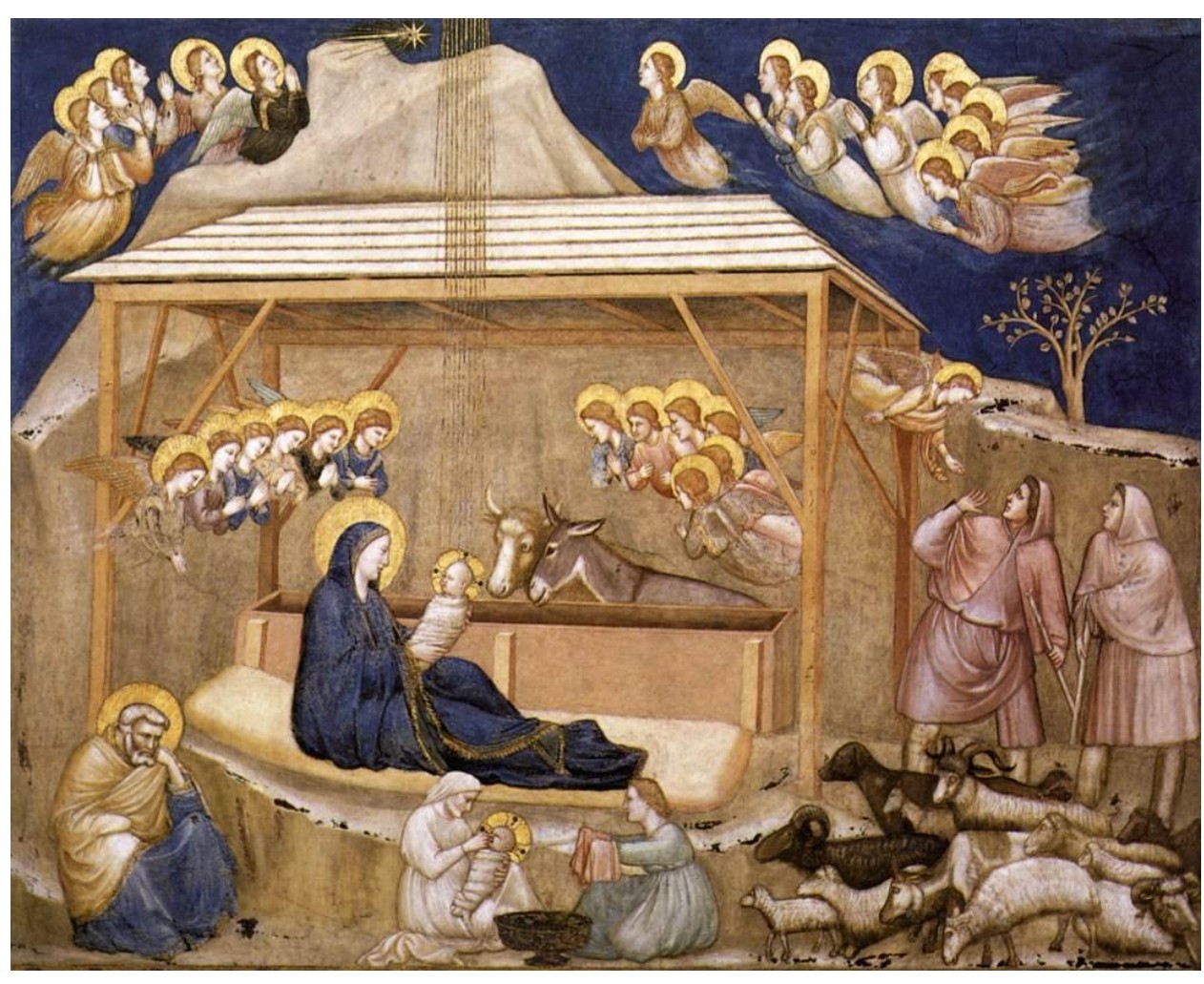


Fig. 12. Giotto. Nativity. Lower Church, Assisi c. 1310. Public domain Wiki Commons

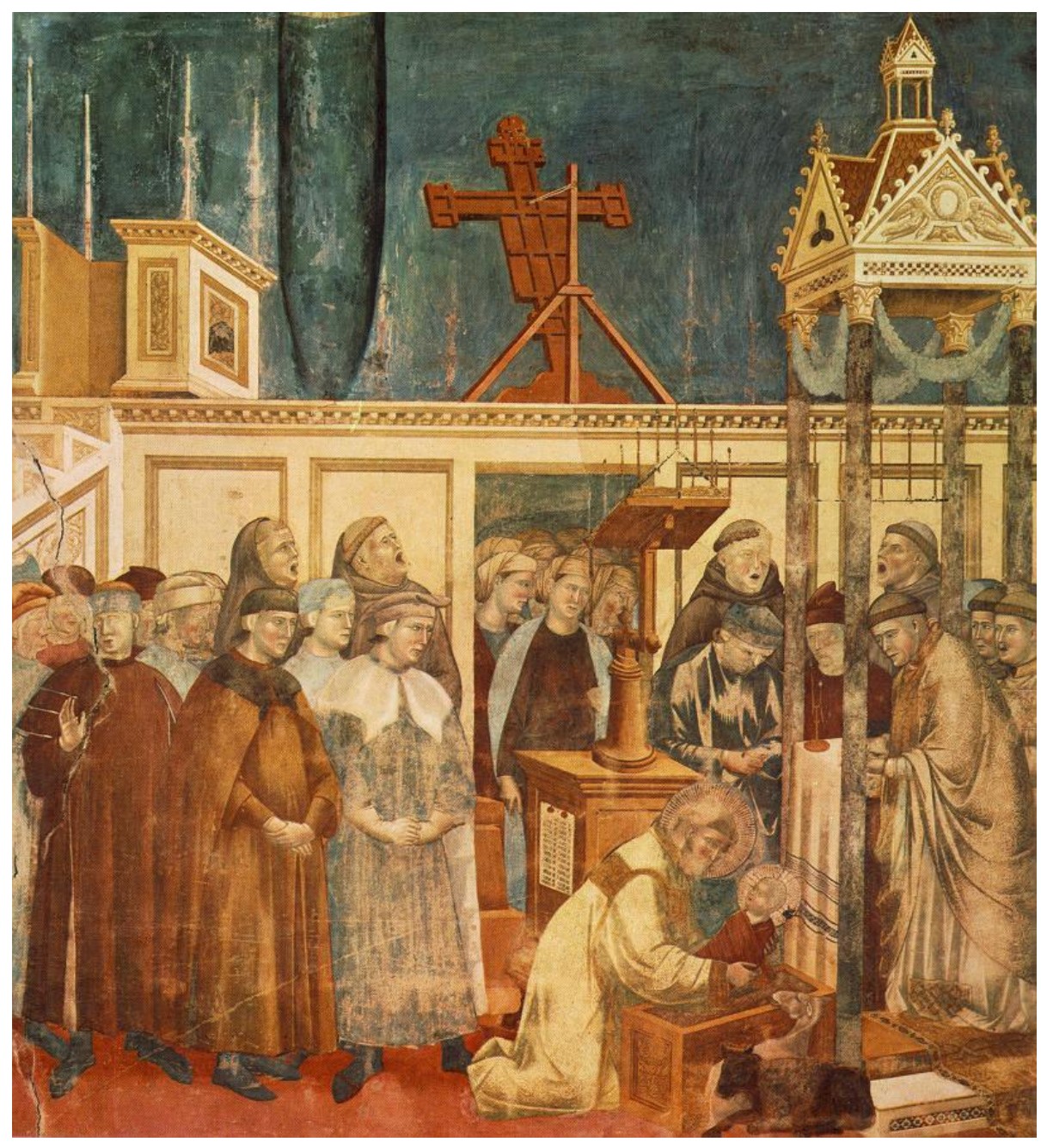



Fig. 13. Giotto. Nativity scene. C. 1297-1300. St. Francis Upper Church Assisi. Note the change of locus from the outdoor manger to
an urban, interior, populated, public church. Public domain Wiki Commons





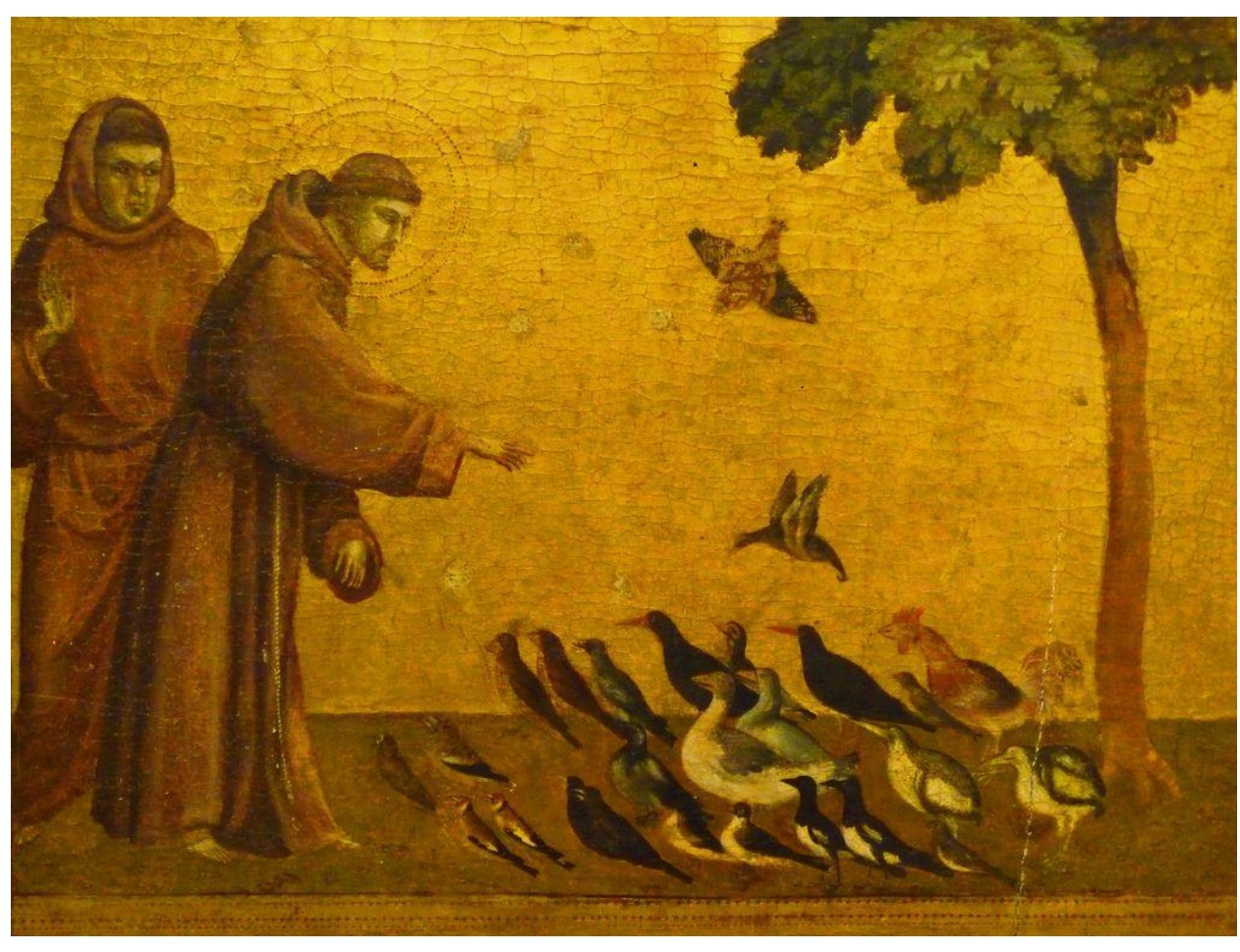

Fig. 14. St. Francis Preaching to the Birds. Giotto. 1295-1300. Louvre. Public domain Wiki Commons

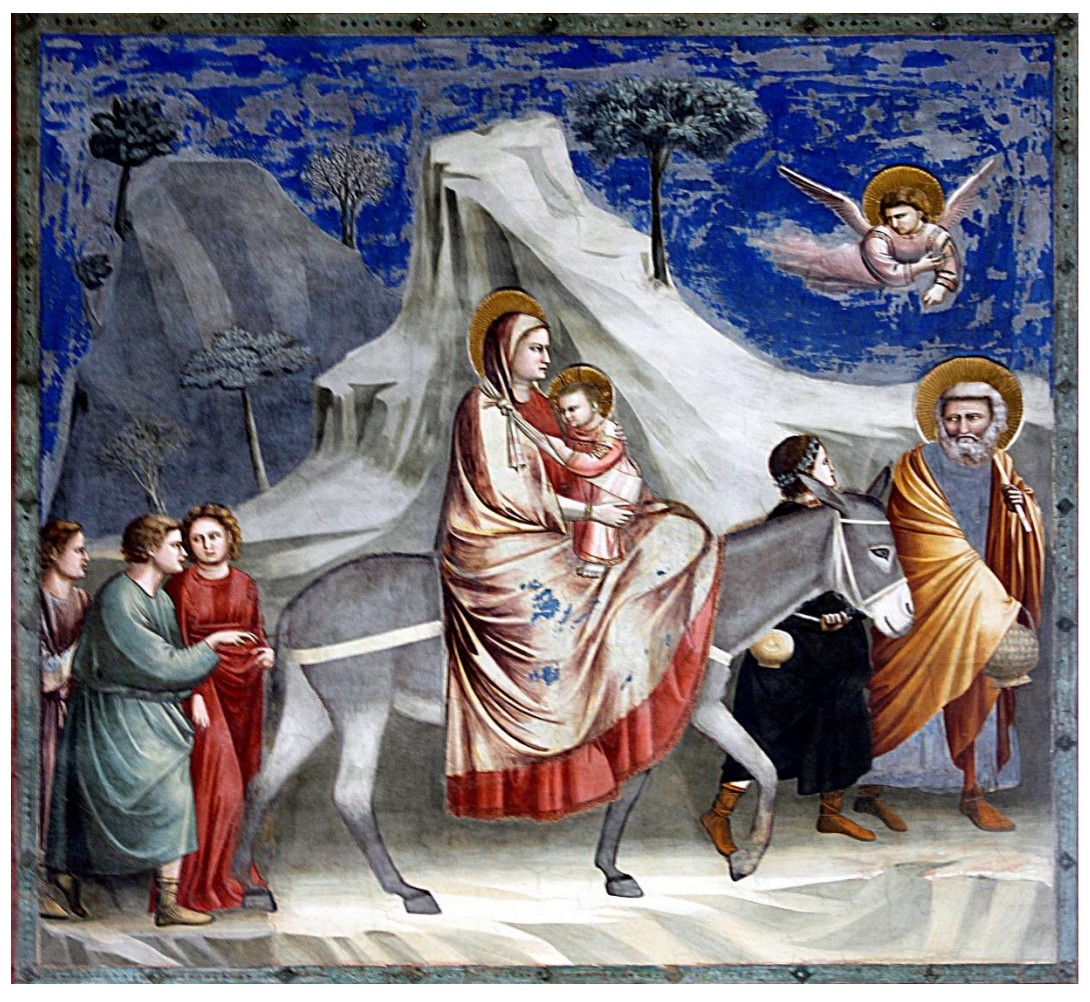


Fig. 15. The Flight into Egypt. Giotto. 1304-1306. Scrovegni (Arena) Chapel, Padua, Italy. Public domain Wiki Commons

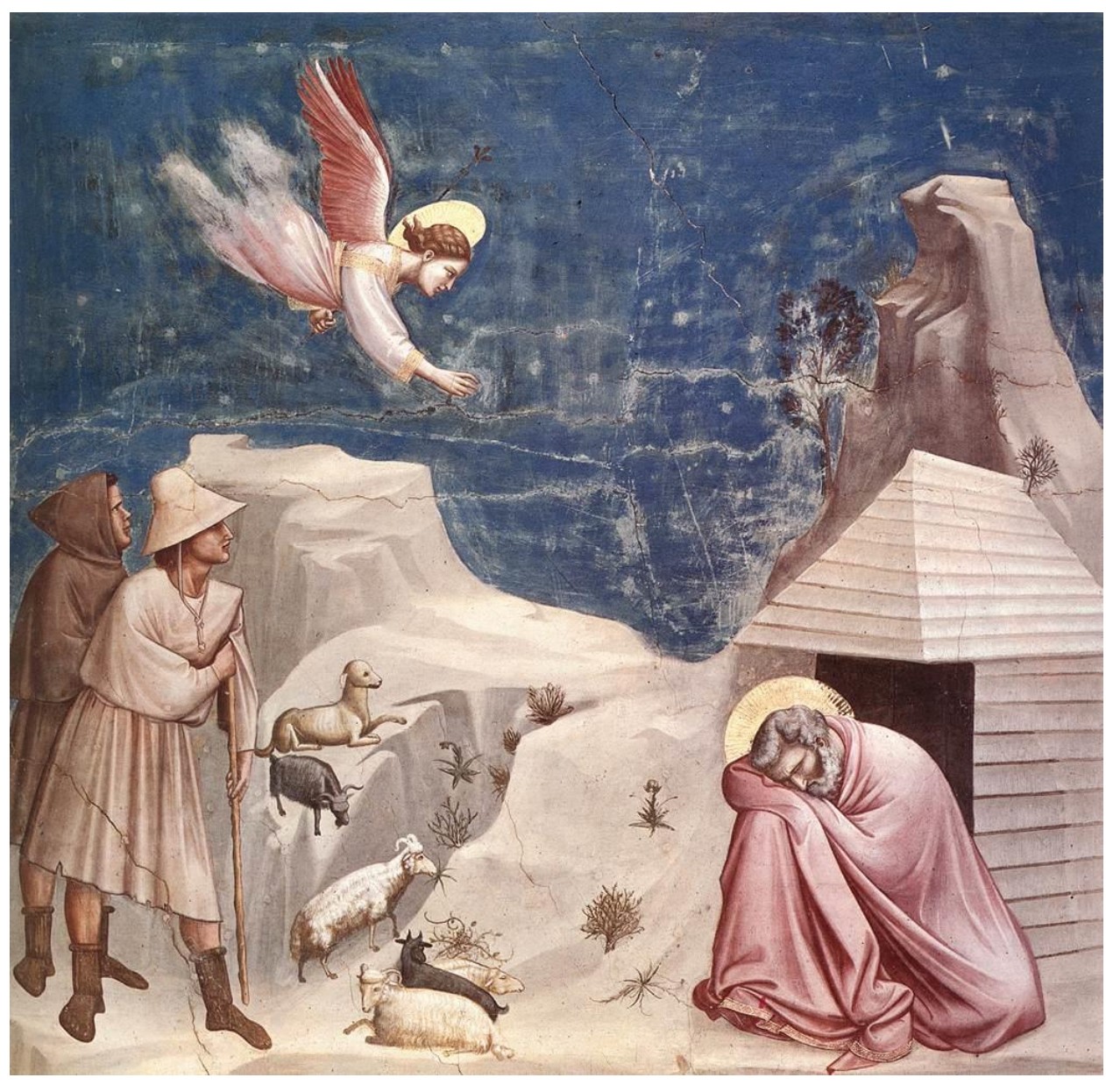


Fig. 16. The Dream of Joachim, Giotto. 1304-06 Scrovegni (Arena) Chapel, Padua, Italy. Public domain Wiki Commons

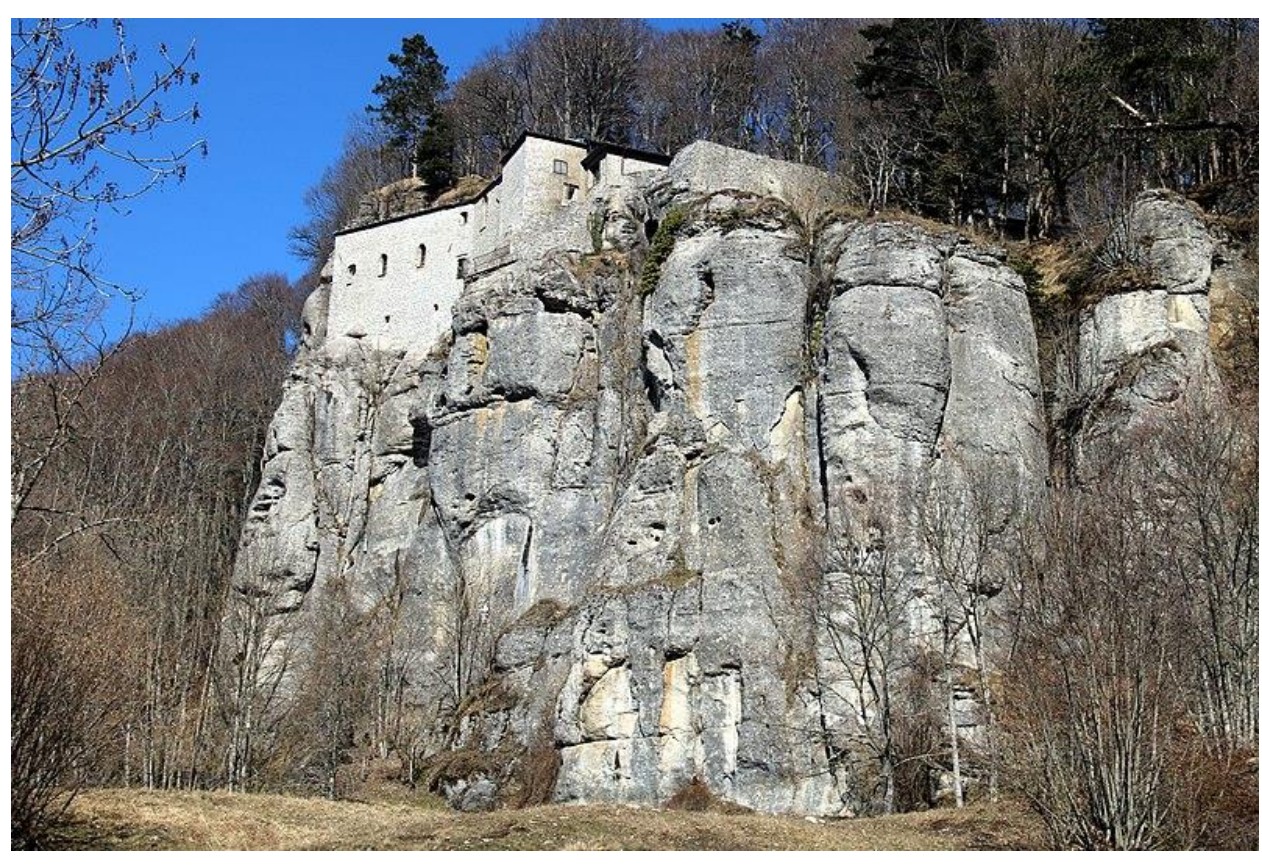


Fig. 17. The monastery at La Verna located on Mt. Penna composed of Miocene calcarenite. Public domani Wiki
Commons

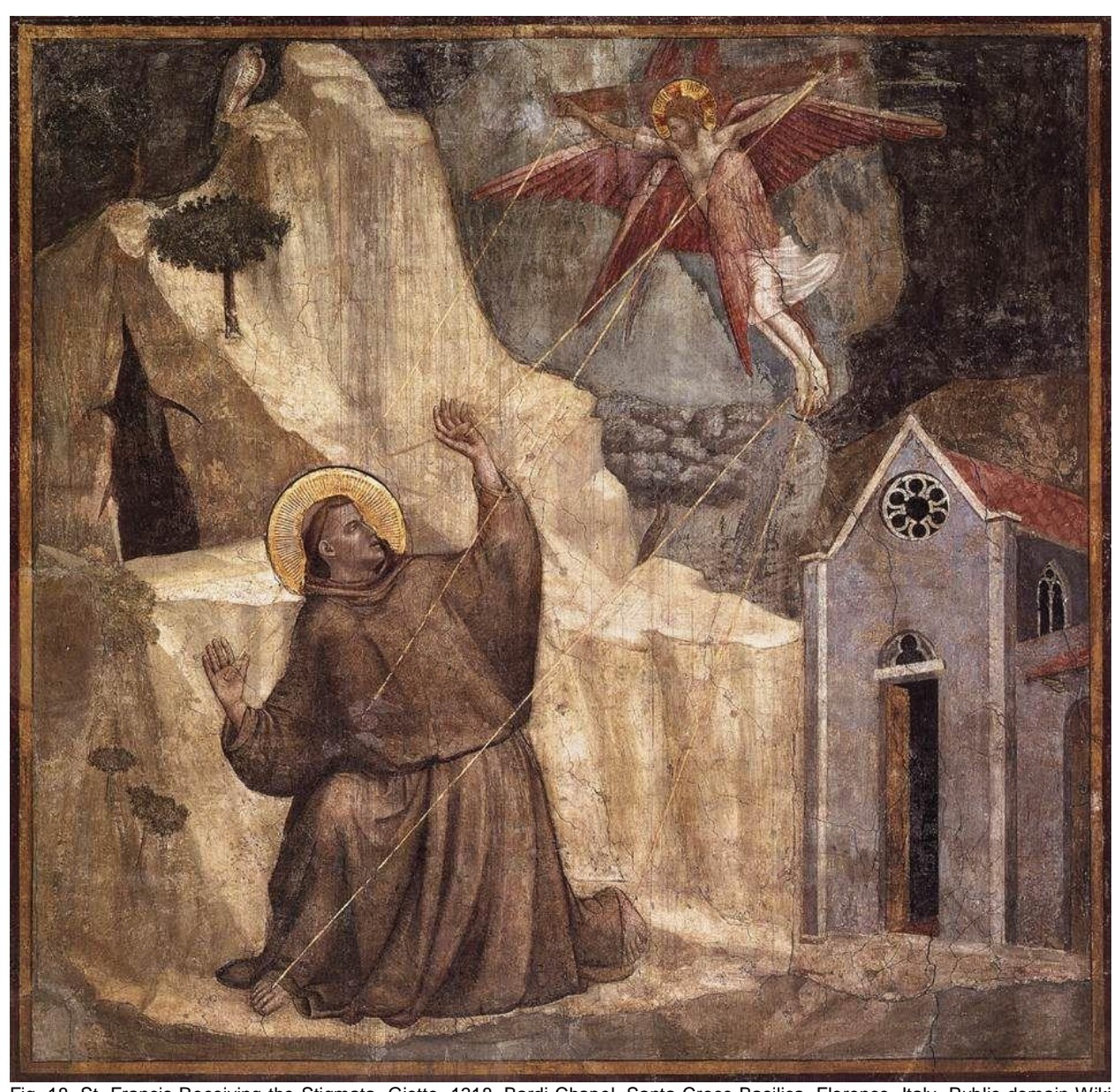

Fig. 18. St. Francis Receiving the Stigmata. Giotto. 1318. Bardi Chapel, Santa Croce Basilica, Florence, Italy. Public domain Wiki
Commons


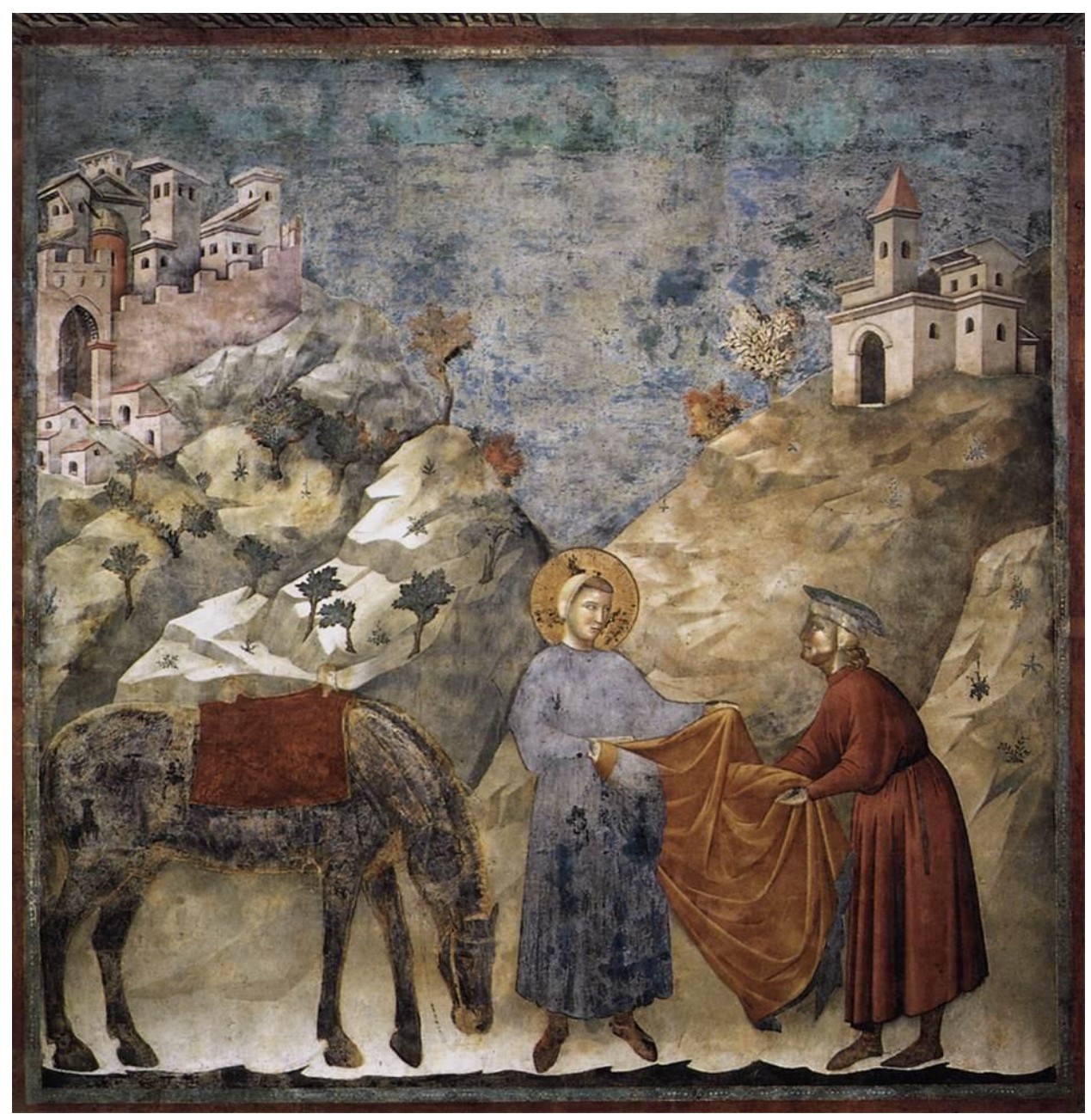


Fig. 19. St. Francis Gives His Mantle to a Poor Man. 1297-1299. Basilica of Saint Francis, Assisi, Italy. Public domain Wiki Commons




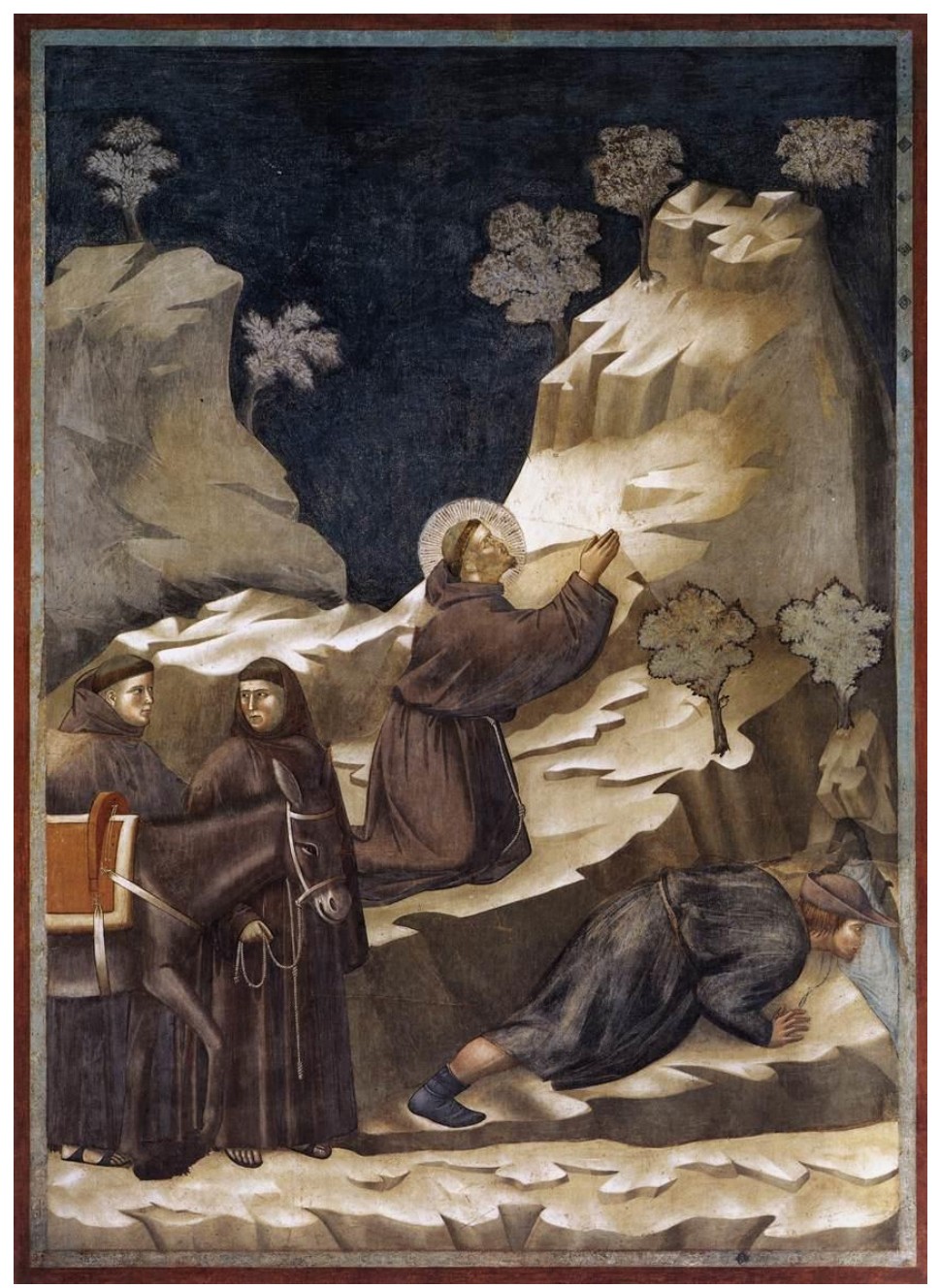

Fig. 20. The Legend of St. Francis: Miracle of the Spring 1297-1300 Upper Church, Assisi. Note the realistic depiction of the wave-like marks of erosion along the bedding planes of the stratified limestone. Public domain Wiki Commons
