# Peer review of "A PORTRAIT OF CENTRAL ITALY'S GEOLOGY THROUGH GIOTTO'S PAINTINGS 2 AND ITS POSSIBLE CULTURAL IMPLICATIONS"

_Geoscience Communication, 2020_

## Referee Comment (RC1) · Martin Bohle (Referee) · 15 Apr 2020

The paper "St. Francis and Giotto: The Saint and The Artist Who Started the Ecological Movement" by Ann C. Pizzorusso builds on the study of paintings of the Italian painter Giotto (c.1270 – 1337) as seen by a geologist. Ann C. Pizzorusso puts these painting in a context of shifting perceptions of man and nature; first, at times of the artist's life (and following centuries), and second, at contemporary times. The paper illustrates communication between the geological reading of artistic presentations of landscapes that were painted seven hundred years ago and contemporary veneration of a famous theological practitioner (St. Francis, 1181/82 – 1226). The strength of the paper is found in the first element, its weakness in the second element. Notwithstanding that this dichotomy must be adjusted for a scientific paper, the cognitive, affective and spir-

itual sense-making that the author communicates "…analyze the rock formations and marvel at the mystery of the Earth's topography, it takes an artist to move me to tears by presenting their unique view of the landscape. In doing so, an artist presents the Earth using the visual—a tool so powerful it can move the most naïve viewer…" (line 42-45) – is a valuable observation; that is worth to communicate, too. Other than geological thinking, this paper requires insights in social science, humanities and arts to consider: (1) why Giotto and other painters have chosen (new) elements of style; (2) how the viewer may have perceived them; and (3) to hypothesize about the cultural impact that may have come from it. This cross-section of various elements makes the paper a fascinating reading - currently, more like an essay than a scientific paper- which motivates the reader to deploy different angles of view. The geological reading of Giotto's paintings allows identifying those features within the geology of central Italy that likely did inspire the artist. The respective parts of the paper (sections 1.6-1.11) are 'rock-solid', although it remains hypothetical what the artist did notice for inspiration. Anyhow, this part of the paper is central to illustrate how geology and arts interact; a subject of which the Ann C. Pizzorusso has compelling experiences. These observations should be moved further to the forefront of the paper. The part of the paper (sections 1.1 and 1.3) that describes shifts in (religious) arts of the thirteens century in junction with shifts in the ecclesiastic hierarchy (Gregor IX – Nicolas IV) would benefit from literature references, in general. Furthermore, it would be beneficial if these references could support the author's conjectures about the modified perception of the thirteens-century-viewer. The given description of the perception of a viewer living in the thirteens century seems probable. Nevertheless, the description may be tainted by projecting modern world-views; a possible bias that should be considered. Therefore, as far as possible, evidence from historical sources should be added that support the conjectures. Depending on the amount of evidence that can be added, this part of the paper should be repositioned compared to the observations that are reported. Sections 1.2, 1.4 and 1.12 may be taken as an account of spiritual respect (veneration). As such, they have little function in a scientific paper. Nevertheless, they illustrate nicely the central hypothesis of the author, namely that 'that Giotto's (and others) artistic style was a powerful communication vehicle'. Notwithstanding that this hypothesis seems valid (and likely can be proven), the author states "[t]he unlikely partnership of St. Francis and Giotto, two revolutionaries, changed Western piety, art history and natural philosophy (line 456-457)". This claim goes a little far for two persons who never met because they lived a generation apart. Such a strong claim – already presented in the title that possibly can be improved - would need correspondingly compelling evidence; a little more than mentioning a political move of an ecclesiastic representative in the twentieth century (line 441 ff). Nevertheless, this political move may be interpreted as identifying St. Francis as a/the pioneer of ecological thinking within catholic traditions. Summarising, sections 1.2, 1.4 and 1.12 should be reviewed in great depth if kept as part of the paper. Overall, the corpus of literature added at the end of the paper seems too limited for the subject. The reader should benefit from a robust corpus of references. The language and the style read well. The essay-like style of the paper is suitable for this piece of scholarly work that is more explorative than affirmative. The paper should be publishable after revision.

---

## Referee Comment (RC2) · Anonymous Referee #2 · 18 May 2020

The thematic content of this paper is very interesting with significant interpretative ideas including, in particular, the innovative concept of Nature identified in Giotto's paintings as an ecological reading of the life of San Francesco. However, the chapters should be reviewed and organized according to the rules of scientific work. 1) A general introduction to the rich bibliography related to Geology and Art (Cultural Geology) would be appropriate. 2) An indispensable chapter on methodology is also missing. The reader should be guided to understand how the comparisons between the paintings and the landscapes have been made. If the method is speculative / intuitive or based on scientific analyses. In they last case would be helpful to refer to geological studies conducted on the area where those landscapes have originated. 3) It is necessary to frame the area from a geological point of view. It would be the case to refer in the text

to the last figures of the paper 18-19-20 and devote a paragraph to the geology of the area even for non-specialists, in a simple but rigorous way. The text completely lacks historical-literary citations on the landscape during the Middle Ages. This prevents the reader from understanding what is said in an original way by the author or "already said" by other researchers.

I consider this paper important and I hope for its publication. For this reason, I recommend the author to follow the previous suggestions and the reviews to the text that I indicate as follows:

30-31 Please note that other regions of Italy are dotted with Franciscan convents, including the Marche and Emilia Romagna

64 The Idea of Nature in the Middle Ages This is a very interesting chapter on the concept of Nature in the Middle Ages, and it would need a reference to one / two fundamental works on this concept.

69 Please add reference for the method called "anagoge"

92 Fig. 1 The map corresponds only to a part of the Franciscan monasteries, as there are famous monasteries built precisely by the will of Francis also in Emilia Romagna and in the Marche regions. Please note that these could also be relevant for the present study.

103 "It was built with indigenous pink Subasio limestone" please provide references

113 – 114 "Giotto's works were so revolutionary that today he is considered the founder of Renaissance Art". Please provide some references.

117-118 "Instead, he made them more realistic by using proportions and shading for volume". Please provide some references.

131 "Interest in perspective had been lost in the preceding centuries". Since when it was lost?

136-140 "the two artists. . .medieval viewer" Can this be a first, shy approach to detail that will emerge only later in the Renaissance with Piero della Francesca?

156-157 I consider this concept very important, and it would be interesting to develop this with respect to the several studies on the topic.

164 We suggest remove the regions (Umbria, Lazio and Tuscany) leaving only "Central Italy".

175 insert the number of the figures.

176 1.4 St. Francis Preaching to the Bird. The reason for the description of this fresco at this point of the paper is not understood. I suggest an appropriate introduction to justify this chapter. Please include the reference to Figure 3

177-179 "In Byzantine art, the background was usually gold, a glorious, expensive color which invoked a sense of awe of the Divine and, as a result, kept the viewer at a reverential distance" Please provide some reference

195-200 I suggest to move these sentences at the beginning of the chapter so it can better understood why this fresco is inserted.

208 Please insert in brakets: (252-201 Mya)

212-214 It would be interesting to know what type of climate was in that period in central Italy.

247 I suggest to remove "violent".

252 From the figures, it is difficult to identify the limestone cliffs, crevasses, and the original grotto especially for non-specialists. Please use some graphics to explain what the reader can see in the pictures.

339 Insert here the figure relative to the painting (fig. ...)

346-347 The geology of this region has been studied and known for a long time. It is

necessary to insert an adequate bibliography easily available on the net. So please make references to some geological works in this sense and indicate which studies indicate that "Mt. Penna (Fig. 13) is composed of Miocene calcarenite resting Cretaceous successions belonging to the eastern Ligurian Units (Sillano Formation, Early Cretaceous).

376 1.10 St. Francis Gives His Mantle to a Poor Man

While in the previous paintings it can be said that the geology present in the backgrounds represents the local one (Greccio and La Verna), in this fresco it is more difficult to say that the small town is Assisi. San Francesco is clearly traveling and this village could also be in areas further away from Umbria (for example in Emilia Romagna, where the morphology of the cliffs is very similar to that of the fresco). The Franciscan convent of Assisi for example is not on a hill in front of the city but inside the inhabited area. A detailed historical research on the places visited by Giotto would be necessary.

389 Where exactly? Can you give examples?

389 - 391 "The gaping…the region" The sentence is a bit confusing especially for a non-geologist since verticalizations are the product of thrusts and activation of faults in a very long period.

407 Insert here (Fig. 16)

413 It seems that they are vertical fractures, in fact they are dip slope layers detected by selective erosion.

416 "the ongoing seismic activity of the area" please give evidences that there were earthquakes in that period.

418 Please note that in this area there are no widespread sinter terraces. (please refer also to what is said for fig. 17, and to what is suggested for line 413)

424-426 Please note that this is not supported by geological data.

478-480 reference not mentioned in the text

481 reference not mentioned in the text

482-483 reference not mentioned in the text

484-487 reference not mentioned in the text 488-489 reference not mentioned in the text

496 497 498 499 references not mentioned in the text

Figures:

Fig 1 please refer to the comment for line 92

Fig. 2 the location of this outcrop is missing

Fig. 3 low resolution figure (I hope it is better in the original)

Fig. 5 low resolution figure

Fig. 7 Caption too long, part of this text may have been inserted in the text.

Fig 8 please correct "lactustrine" in "lacustrine"; "Greccio sits in the area..." Do you mean the inhabited Greccio or the Convent? It is not clear; mistake in the limestones dating

Fig 9 Please correct "Assissi" in "Assisi"

Fig. 11, 14, 15 Low resolution images

Fig. 16 see text about sinter terrace (see references to line 418 and 413).

Fig. 17 I suggest not to put this figure as far as I say in the text (see references to line 418).

Fig. 18 and Fig 19 Fig 20 I suggest making shorter captions and explaining geology in

the text. Fig. 18, 19, 20 does not appear in the text

---

## Author Comment (AC1) · 30 Jun 2020

Thank you. I have incorporated your valuable and much appreciated comments into the final text. Your time and effort are very much appreciated.

---

## Author Comment (AC2) · 30 Jun 2020

Thank you. I have incorporated your valuable and much appreciated comments into the final text. Your time and effort are very much appreciated.

---

## Editor Decision (ED2)

**Final corrections**

Line 518: I suggest to delete "the power of"

Line 661 remove italics

Please note that Fig. 5 and Fig. 6 are not of good quality. Discuss it with the Editorial support.

Thank you.

[revised manuscript text omitted]